

# Preferential Flow Systems Amended with Biogeochemical Components: Imaging of a Two-Dimensional Study

Ashley R. Pales[1], Biting Li[1], Heather M. Clifford[2], Shyla Kupis[1], Nimisha Edayilam[3], Dawn Montgomery[1], Wei-zhen Liang[1], Mine Dogan[1], Nishanth Tharayil[3], Nicole Martinez[1], Stephen Moysey[1], Brian Powell[1], and Christophe J. G. Darnault[1*]

[1]Department of Environmental Engineering and Earth Sciences, Laboratory of Hydrogeoscience and Biological Engineering, L.G. Rich Environmental Laboratory, Clemson University, 342 Computer Court, Anderson, SC 29625, USA.
[2]Climate Change Institute, University of Maine, Edward Bryand Global Sciences Center, Orono, ME, 04473, USA.
[3]Department of Plant and Environmental Sciences, Clemson University, Clemson, SC, 29634, USA.

Corresponding to Christophe J. G. Darnault (cdarnau@clemson.edu)

**Abstract.** The vadose zone is a highly interactive heterogeneous system through which water enters into the subsurface system by infiltration. This paper details the effects of simulated plant exudate and soil component solutions upon unstable flow patterns in a porous media (ASTM silica sand; US Silica, Ottawa, IL, USA) through the use of two-dimensional (2D) tank light transmission method (LTM). The contact angle and surface tension of two simulated plant exudate solutions (i.e. oxalate, and citrate) and two soil component solutions (i.e. tannic acid, and Suwannee River Natural Organic Matter) were analyzed to determine the liquid-gas and liquid-solid interface characteristics of each. To determine if the unstable flow formations were dependent on the type and concentration of the simulated plant exudates and soil components, the analysis of the effects of the simulated plant exudate and soil component solutions were compared to a control rainwater solution. The differences in the fingering flow were quantified with the finger geometries, the velocity of finger propagation, the vertical and horizontal water saturation profiles, and the water saturation at the fingertips. Significant differences in the interface processes indicated a decrease between the control and the plant exudate and soil component solutions tested; specifically, the control at 64.5 θ and 75.75 Nm/m, to the low concentration of citrate at 52.6 θ and 70.8 Nm/m. The changes in finger geometries and velocity of propagation between the control solution and the simulated plant exudate and soil component





solutions further demonstrate that the plant exudates increased the wettability and mobility of the solutions during the infiltration process in unsaturated porous media.

**Keywords:** Infiltration; Wetting front instability; Fingered flow; Interface phenomena; Plant
exudates; Soil constituents; Light transmission method

## 1. Introduction

Infiltration of water in soil is a fundamental element of the hydrologic cycle (Horton, 1933). The infiltration process controls the existence of water and life at the soil-atmosphere interface, especially soil moisture (Rodriguez-Iturbe et al., 1999) and vegetation (Liang et al., 1994).
Consequently, the knowledge of water flow and contaminant transport through soils is critical for the management and development of groundwater resources. Understanding the movement of water in soils is particularly valuable in terms of groundwater recharge and contaminant loading. The infiltration process is impacted by the physico-chemical properties and biological activities that occur in the unsaturated zone of the soil (vadose zone), as well as the human activities and
the climate (Glass et al., 1991).

In general, water flow and contaminant transport in soil are heterogeneous (Bien et al., 2013; Dorr and Munnich, 1991; Klos et al., 1999; McBride, 1998; Muller and Prohl, 1993; Wang et al., 2006). These heterogeneities can occur in the form of spatial and temporal variability in soil
hydraulic properties, water supply rate and contaminant amount. These factors affect the hydrologic response to precipitation, and thus, influence the quantity and quality of both groundwater and surface water resources (Beven and Germann, 2013; Clothier et al., 2008; Jarvis et al., 2016; Jarvis, 2007; Weiler, 2017). Preferential flow, is one such heterogeneity, which is the rule rather than the exception in hydrologic and earth sciences (Bundt et al., 2000;
Smith and Elder, 1999). Preferential flow in soil is defined as the concentration of flow of water through a small fraction of the volume of the porous media, and bypassing most of the soil matrix. It increases the flow velocity (Hendrickx and Flury, 2001) and thus reduces the time for the attenuation of contaminants, and quickens the transport of contaminants to aquifers. Preferential flow, has been the subject of numerous studies (e.g. (Allaire et al., 2009; Andreini





and Steenhuis, 1990; Clothier et al., 2008; Flury et al., 1994; Gerke et al., 2010; Glass et al., 1989a, 1989b, 1989c, 1989d; Kung, 1990a; Ritsema et al., 1993; Simunek et al., 2003)). Preferential flow in the soil may occur through several mechanisms, such as macropore flow (e.g., connected and disconnected macropores, cracks, earthworm burrow, root channels) (Beven

and Germann, 2013; Bouma and Dekker, 1978), fingered flow (i.e., wetting front/gravity-driven instability or "fingers") (Glass et al., 1989c, 1989d; Wang et al., 2003a), and funnel flow (e.g., layering or textural interfaces) (Kung, 1990a, 1990b). Preferential flow serves as the catalyst for the rapid transport of solutes and particles, including contaminants such as: pesticides, nutrients, metals, pathogens, radionuclides, antimicrobials, and nonaqueous phase liquids (NAPLs); with

little interaction with the soil matrix, significant degradation of groundwater quality can occur (Boyer et al., 2009; Darnault et al., 2003, 2004; DiCarlo et al., 2000; Engelhardt et al., 2015; Jarvis et al., 2016; Kay et al., 2004, 2005; Kim et al., 2005; Nimmo, 2012; Uyusur et al., 2010, 2015; Zhang et al., 2016).

The stability of an interface between two immiscible fluids in a porous medium has been the subject of numerous research since the 1950's until now, especially in the field of petroleum engineering and in hydrologic sciences (Chouke et al., 1959; Homsy, 1987; van Meurs, 1957; Saffman, 1986; Saffman and Taylor, 1958; Stokes et al., 1986; Tanveer, 1987a, 1987b). When a wetting fluid comes in contact with a dry porous medium, flow in the porous medium is caused

by capillarity, and these capillary forces transfer water in any direction. The development of this flow without the influence of gravity is known as a "sorption" process, and the development without this influence is known as an "infiltration" process (Assouline, 2013). Infiltration process in soil is the particular instance of the dynamics of fluid interfaces and flow in porous media, where infiltration is a downward movement of water into dry unsaturated soils and causes

the formation of a gas-water interface, known as the "wetting front". This wetting front propagation can be classified as stable or unstable depending on the non-linear behavior of the relationship between the soil hydraulic properties and the soil moisture content (Cueto-Felgueroso and Juanes, 2008, 2009a, 2009b, 2009c; Egorov et al., 2003; Wallach et al., 2013). During a stable infiltration process into a homogeneous porous media under a steady state flux

condition at the surface, a stable wetting front moves downward. While in an unstable infiltration process into a homogeneous porous media under steady state flow, an unstable wetting front is





driven by gravity and stabilized by the surface tension resulting from capillary forces present in the porous media, thus the preferential flow paths ("fingers") are generated (Glass et al., 1989a, 1989b, 1989c; Hill and Parlange, 1972; Kapoor, 1996; Peck, 1965; Selker et al., 1992a, 1992b; Yao and Hendrickx, 1996). Once these fingered flow phenomena are generated, the flow area is

decreased compared with that of the wetting front, and the flux velocities of the fingers increase tremendously (Wallach et al., 2013). Peck (1965) was the first to describe the formation of "tongues" of infiltrating water in sand columns and the entrapment of air below the wetting front. Hill and Parlange (1972) conducted the first systematic experiment to investigate the phenomenon of wetting front instability. The role and importance of fingered flow in

contaminant transport was demonstrated with several cases of groundwater contamination (e.g. (Hillel, 1987; Steenhuis et al., 1990)).

The non-linearity of the governing equations for unsaturated flow through porous media makes a theoretical analysis of water movement in soils and wetting front instability in soils most

challenging (Buckingham, 1907; Richards, 1931). Infiltration theories and the development of mathematical solutions to solve the problem of flow in porous media have been the subject of much research, most particularly as evidenced through the seminal work of Jean-Yves Parlange (Parlange, 1971a, 1971b, 1971c, 1972a, 1972b, 1972c, 1972d, 1972e, 1973, 1975; Parlange and Aylor, 1972). Analytical and quasi-analytical solutions of the flow equation have been developed

through the time expansion (Philip, 1969, 1975) and integral approaches (Parlange, 1971b, 1972e). Further Saffman and Taylor (1958) and Chouke et al., (1959) concluded that the development of wetting front instability resulted from unsaturated flow (Pendexter and Furbish, 1991; Raats, 1973; Selker et al., 1992c), an increase in hydraulic conductivity with depth (Hill and Parlange, 1972; van Ommen et al., 1989), a decrease in soil wettability with depth (Hill and

Parlange, 1972; Hillel, 1987; Philip, 1975; Raats, 1973; Tamai et al., 1987), water repellency (Dekker and Ritsema, 1994; Ritsema et al., 1993; Ritsema and Dekker, 1994), a redistribution of infiltration after the end of rainfall or irrigation (Philip, 1975), and air entrapment (Peck, 1965; Philip, 1975; Raats, 1973; Wang et al., 1998).

In addition to these theoretical analyses, numerous experimental studies have been undertaken to elucidate the instability of the wetting front propagation, specifically in layered soils (Hill and





Parlange, 1972), under increased air pressure ahead of the wetting front and soil heterogeneity (White et al., 1976, 1977), and in layered soils with variable initial soil moisture (Diment and Watson, 1985). These conditions are prevalent in soils through the world. The instability of wetting fronts has been studied primarily in two-dimensional tanks entailing the generation of a

two-dimensional fingered flow (e.g. (Baker and Hillel, 1991; Bauters et al., 2000a; DiCarlo, 2004; Glass et al., 1989a, 1989c, 1989d, Selker et al., 1992a, 1992b, 1992c)). Laboratory experiments designed to examine the infiltration process in homogeneous porous media have confirmed the constant formation of fingered flow (Bauters et al., 2000b; Diment and Watson, 1985; Glass et al., 1989d; Lu et al., 1994; Selker et al., 1992a; Sililo and Tellam, 2000; Wang et

al., 2003b, 2004; Yao and Hendrickx, 2001). Also experimental observations have demonstrated that the fingers that develop the most rapidly will convey most of the flow, while the other initial fingers will be quenched (Glass et al., 1989d; Selker et al., 1992b). The fully formed fingers generated by gravity-driven flow behave like traveling waves (i.e. constant shape and velocity) and the water saturation profile of the fully form fingers exhibit the following pattern: a region of

high water saturation directly behind the wetting front, known as the fingertip, i.e. an overshoot at the fingertips; followed by another region of drainage where the water content in the finger decreased to a lower and uniform water saturation, known as the finger tail (DiCarlo, 2004; Glass et al., 1989a; Liu et al., 1994; Selker et al., 1992b). Further details on theoretical and experimental research on wetting front instability can be found in extensive reviews (Assouline,

2013; Glass and Nicholl, 1996; De Rooij, 2000; Xiong, 2014).

The vadose zone is a highly interactive heterogeneous system between water, air, soil constituents, flora, and fauna (Bengough, 2012; Hallett et al., 2013; Lehmann et al., 2012). The rhizosphere is an important component of the vadose zone and is specifically defined as the

narrow region of soil that is directly influenced by plant roots themselves and by the root secretions (Hinsinger et al., 2009; McCully, 1999; Passioura, 1988; Read et al., 2003). It has been estimated that about 40% of the precipitation on earth transitions through the rhizosphere (Bengough, 2012). The rhizosphere has a deep effect on the hydrologic processes (Benard et al., 2016), and its interactions in the soil system not only involve the physical effects stemming from

soil and root dynamics, but also include biochemical effects from the compounds the roots of a plant can exude from their root tips (Benard et al., 2016; Gregory, 2006; Hinsinger et al., 2009).



There are two main biochemical compounds produced by the plant roots: mucilage and exudates. Mucilage is a viscous substance composed mainly of polysaccharides and some phospholipids (Guinel and McCully, 1986; McCully and Boyer, 1997; Read et al., 2003) the purpose of which
is to keep the soil around the roots moist by swelling and adsorbing water (Guinel and McCully, 1986), as well as to control the water repellency of the rhizosphere (Carminati, 2012; Carminati et al., 2010; Moradi et al., 2012). Root exudates comprise mostly sugar, amino acids, and organic acids, and are exuded primarily from the root tips (Carvalhais et al., 2011; McCully, 1999; Passioura, 1988; Read et al., 2003). These exudates play a key role in keeping the contact
between the roots and soil particles (Walker et al., 2003), the modulation and mobilization the nutrients (Cakmak et al., 1998; Carvalhais et al., 2011; Wang et al., 2008), and the maintenance of the rhizosphere hydraulic properties (Ahmed et al., 2014) by providing a large water-holding capacity (McCully and Boyer, 1997), enhancing the movement of water in dry soil, and enabling the roots to get water from dry soils (Ahmed et al., 2014). Individual exudates have specific
nutrients (i.e. phosphate, nitrate, potassium) that they are designed to extract from the soil. This unique chemistry thus increases the solubility of the sought-after nutrients to ease the uptake by the plant. In this regard, the exudates act as surfactants within the subsurface by lowering the surface tension and contact angles of the soil pore water, enhancing the wettability, and reducing the suction and energy needed for the plant to extract what is needed (McCully, 1999; Passioura,
1988; Read et al., 2003).

The plant root growth in soil is constrained, however, as water, air, and nutrient resources are scarce, and spatially and temporally heterogeneously available in soil (Hinsinger et al., 2009), plants have evolved various adaptive strategies to adapt to the environment they encounter to
their advantage and access these resources. Hydromechanic analyses of soil and water interactions in the rhizosphere have demonstrated that rhizodeposits (i.e. mucilages and exudates) induce water potential gradients near the roots and therefore increase soil moisture near them (Carminati et al., 2010; Gardner, 1960; Ghezzehei and Albalasmeh, 2015; Moradi et al., 2011; Young, 1995). As the water potential decreases near the root to produce a concurrent
decrease in soil water content, water moves from the bulk soil to the root surfaces. Although the effect of root exudates on the soil structure and aggregation has been well established (Alami et



al., 2000; Czarnes et al., 2000; Ghezzehei and Albalasmeh, 2015; Kaci et al., 2005; McCully, 1999; Watt et al., 1994), research on the effect of the rhizodeposits (i.e. mucilages and exudates) on soil-water relationship and water dynamics in the rhizosphere is very limited. This is a crucial knowledge gap in terms of the biogeochemical influences on the hydrologic processes occurring

in the rhizosphere as stated by Bengough (2012), particularly the effect that root exudates have on the nonequilibrium dynamics of interfaces and flow in porous media. Carminati et al. (2016) recently analyzed the rhizosphere processes to more thoroughly elucidate the system, with a particular focus on determining the role of mucilage in root water uptake. They were particularly interested in determining the biological impacts of wet versus dry mucilage to the system, and

how those changes in turn, affected the plants overall ability to uptake water (Carminati et al., 2016).  They found that wet mucilage enhances fluid transport, but when dry causes hydrophobic tendencies in the rhizosphere (Carminati et al., 2016).  From this analysis they determined a need for new technology and sensor applications to gain a full and detailed understanding of the rhizosphere, regardless of the new methodologies in development for imaging these systems and

understanding the plants themselves. Ultimately an enhanced focus in current research to better elucidate the interactions between the biological, chemical, and physical processes that control the rhizosphere is needed. A knowledge of the mechanisms and parameters that govern the wetting front dynamics is most necessary for predicting wetting-front propagation through soil, which in turn makes it possible to predict both soil and groundwater pollution. The non-uniform

nature of transport occurring through unstable flow fields means that the transport of contaminants in the subsurface and contaminant loading of aquifer differs from an assumed one-dimensional modality of movement (Glass et al., 1989b, 1991). Any contamination to the vadose zone contamination means a threat to groundwater, which also means that any remediation undertaken to remove that threat is most challenging (Dresel et al., 2011; Triplett et al., 2010;

Wellman et al., 2010).

Indeed, the groundwater at the decommissioned nuclear production Hanford facility in  Richland, WA is contaminated with uranium (Qafoku et al., 2004; Rod et al., 2010; Um et al., 2009, 2010) caused from discharge from leaky underground storage tanks of more than 7 tons of uranium

through the vadose zone in an alkaline milieu (Jones et al., 2001). This leakage, which occurred from the late 1950s through the 1960s resulted in groundwater contamination in parts of the 300



Area Hanford Site that surpasses the 30 $\mu gL^{-1}$ uranium drinking water standard (Waichler and Yabusaki, 2005). Another example of such worrisome contamination is the Nevada Test Site (NTS) in which large quantities of uranium were in the vadose zone, from the approximately 600 underground conducted during its operation (Tompson et al., 2006).

Therefore, the intent of the research described here was to investigate the effects of biochemical compounds from root plants and soil constituents on the spatial and temporal infiltration process, particularly the wetting front stability and fingered flow in a solid-liquid-gas system. The dynamics of interfaces in porous media—gas-liquid and solid-liquid interfaces—under the

impact of various biochemical compounds were monitored through interfacial tension and contact angle measurements. Utilizing the imaging technique of the light transmission method (LTM) that allows the visualization of fluid content and flow in porous media with high spatial and temporal resolution (Darnault et al., 1998, 2001; Niemet and Selker, 2001; Weisbrod et al., 2003), we studied the effects of several isolated organic compounds, typically found in the

rhizosphere, that were dispersed as artificial rainfall and infiltrate under unsaturated conditions through a homogenous sand porous media on the wetting front stability and fingered flow phenomena in two-dimensional tank. Ultimately, our understanding of the influence of these isolated plant exudates and soil constituents upon these infiltration processes and water saturation distribution within the porous media is a significant step forward to fully elucidating

the range of interactions controlling flow within the rhizosphere.

## 2. Materials and Methods

### 2.1 Simulated Constituent Solution Preparation

In order to reduce the number of active variables present in this work, solutions of simulated plant exudates and soil components were used rather than living plants and soil. A Hoagland

Nutrient Solution (HNS) with 0.01 M NaCl (NaCl+HNS) was used as a control to simulate the requisite nutrients required of a plant to survive in a hydroponics system. The HNS was created by combining the components listed in Table 1 with 970 mL of ultrapure deionized (DI) water (18.2 MΩ•cm, Millipore Corporation, Billerica, MA) with NaCl (Lot #: 2954C512, VWR Analytical BDH®, Radnor, PA) subsequently added to bring the solution to 0.01 M NaCl. These



steps are based on the method developed by Arnon and Hoagland in 1940. Two plant exudates, sodium citrate (S1804-500G, Sigma-Aldrich, Lake Cormorant, MS), and sodium oxalate (Lot #: P17A014, Alfa Aesar, Ward Hill, MA); and two soil components, Suwanee River Natural Organic Matter (SRNOM, RO isolate, 2R101N, International Humic Substances Society, St.

Paul, MN), and tannic acid (Lot #: MKBV0516V, Sigma-Aldrich, Lake Cormorant, MS), were used to create the solutions tested in this work. Using NaCl+HNS as a base solution, different concentrations of the constituents were created to simulate a range of potential concentrations, a list of which is provided in Table 2. These concentrations were categorized based on amounts naturally found in the environment (Jones, 1998; Strobel, 2001; Wagoner et al., 1997).

**2.2 Solution Characterization**

A Kruss Easy Drop (FM40Mk2, Kruss GmbH, Germany) was used to obtain the contact angle and surface tension with 5 µL of the various experimental solutions deposited on a 24 × 60 mm micro cover glass (CAT No. 4404-454, VWR, Radnor, PA) for the contact angle analysis. The Drop Shape Analysis software (Kruss GmbH, Germany) was used with the following settings:

contact angle profile defined with circle fitting, drop type – sessile drop, sub-type normal. Images and data were collected every 30 seconds for 3 minutes for both the contact angle and surface tension data. The Kruss Easy Drop DSA1 internally computes the contact angle of a drop using the Young equation (Young, 1805):

$$\sigma_s = \gamma_{sl} + \sigma_l * \cos(\theta) \tag{1}$$

where $\sigma_s$ is the solid surface tension, $\sigma_l$ is the liquid surface tension, $\gamma_{sl}$ is the interfacial tension between the phases, and $\theta$ is the contact angle. It also calculates the surface tension of pendent drops using the Young-Laplace equation (Laplace, 1806; Young, 1805):

$$\Delta P = \sigma * \left(\frac{1}{r_1} + \frac{1}{r_2}\right) \tag{2}$$

where $\Delta P$ is the difference in pressure of the inside and outside of the droplet, $r_1$ and $r_2$ are the

principle radii of the droplet, and $\sigma$ is the surface tension. By using the characteristic shape and size of the droplets, the equipment can determine the surface tension of the droplet. These variables on a stable solution droplet collected at 90 s are illustrated in Figure 1 with all solutions tested in triplicate with the averages, and associated standard deviations presented in this work (Table 3). The surface tension measurements were obtained by suspending the solution as a drop





in an open-air environment from a 2 mm diameter needle point. The software was configured with the following settings: drop information – with a needle diameter of 2 mm, a tip level (pixels) of 80, an assumed liquid drop density phase of $(g/cm^3)$ - 0.9975, an embedding phase of 0.0012, a magnification factor (pixels/mm) of 79.827, and an aspect ratio of 1. A statistical

analysis was done for the contact angle and surface tension measurements utilizing a t-test using JMP Pro 12 from SAS.

**2.3 Light Transmission Method**

Developed by N.T. Hoa, the Light Transmission Imaging Method (LTM) was first used as a rapid, inexpensive method for the visualization of flow in 2D systems (Hoa, 1981), and has since

been expanded for use in water-air, oil-water, and water-oil-air systems. The methods used in this work adhere to the methods developed by Darnault et al. (2001, 1998). The LTM works through the illumination of a thin section of porous media (i.e. 2.5 cm thick section of silica sand) from behind with a light source. A standard camera is then used to capture the amount of light that transmits through the 2D area for any given condition (unsaturated – fully saturated).

These images are recorded in standard RGB (red-green-blue) format, and converted into HSI (hue-saturation-intensity) format. The colors of an image or photograph can be broken down into three distinct layers of the image itself, this is the hue, saturation, and intensity. The hue is the wavelength of the light energy, the saturation is the bandwidth of the wavelength, and the intensity is the total energy value or the amplitude. Upon conversion into this format,

information on the water content is discernable. Specific combinations of the HSI format can be used to determine the water content depending on the nature of the 2D system and fluids. Similarly, the porous media is calibrated via saturation to a known percentage and a discrete value is obtained via the HSI format. The experimental design reduced the efficacy of hue and saturation as opposed to intensity; here intensity was used to create the calibration equation for

the experiments.

The 2D tank was constructed using clear 1.3 cm thick polycarbonate plates (8707K153, McMaster-Car, Douglasville, GA) for purposes of visualizing the flow process (Figure 2). The 30.5 × 1.3 × 30.5 cm tank contains four porous plates (0660X01-B01M3, Soilmoisture

Equipment Corp., Goleta, CA) along the sloped bottom of the tank, and an effluent port at the




base of the slope. For experiments, the tank is fitted to an aluminum stand with an LED light source mounted on the back (36W dimmable, natural white 61 × 61 cm, LPWD-NW6060-36, superbrightleds.com, St. Louis, MO) and rainfall simulator (screw actuator) mounted on top. To prevent reflective and refractive light from affecting the images collected, the excess light from

the LED light was blocked to ensure the transmission of only visible light through the tank.

An empirical calibration curve was created using small calibration tanks (7.6 × 1.5 × 8.9 cm) filled with the silica sand to multiple known water saturation percentage points. The water saturation points were found for 0%, 5%, 10%, 20%, 40%, 60%, 80%, and 100%. The

calibration was completed in triplicate to account for potential variations in packing and ambient light pollution. A linear calibration relationship is seen between the water saturation points increasing, and the transmitted intensity increasing (Figure 3), this trend is also observed in the work of Uyusur et al. (2016). It is important to note that the 80% intensity values read high due to water drainage, and thus saturation, of the water in the lower half of the calibration tank. This

linear relationship and slope equation were used to convert the intensity values into water saturation values with the Matlab (R2016a, The Mathworks Inc., Natick, MA, USA). Additionally, for each flow experiment, the 0% and 100% intensity values were found and the slope from each experiment was compared to the calibration curve to ensure that the calibrations between experiments were similar.

**2.4. 2D Tank Flow Experimental Design**

The 2D tank was secured to the stand prior to packing to minimize jostling and to reduce sand movement, thus decreasing the likelihood of heterogeneity. A packing device was then affixed to the top of the tank and filled with two tank volumes of graded sand that conform to ASTM (American Society for Testing and Materials) C778 standards (US Silica, Ottawa, IL). The

second tank volume of sand allows for the homogeneous packing of the tank to a uniform bulk density of 1.5043 g/cm$^3$. The US Silica product data sheet described the size of the ATSM graded sand using standard USA STD sieve mesh sizes from 16 – Pan (US Silica Product Data Sheet, 1997) (Figure 4). The packing device and extra sand were removed from the top of the tank, which was then gently tapped to create an even sand surface. A cover was secured around

the apparatus to block the refractive and reflective light from the LED source, and to ensure that





only transmitted light is seen in the images. Specifically, 1 cm of the top portion of the tank and sand was covered; this dimension is assumed as part of the uniform wetting front for the data presented here.

All tank flow experiments were done in replicate, with data presented in this work derived from the averages of the replica, and the images presented from a single experiment representing the system's characteristics. The secondary experimental images are available for viewing in the appendix. The experimental exudate and soil component solutions were individually pumped into the 2D tank per each experiment. A Cole Parmer peristaltic pump was use for this process
(pump head; Masterflex easy loader II L/S model 77200-60, Masterflex peroxide-cured silicone tubing L/S 16, Cole Parmer Instruments Co., Vernon Hills, IL) with the solutions distributed across the top of the 2D tank by artificial rainfall (rotating screw actuator) over a 45.72 cm$^2$ area, at a constant pump rate of 10 mL/min (flow velocity of 13.12 cm/hr).

A Nikon CMOS D5500 DSLR camera with AF-S DX NIKKOR 18-55mm f/3.5-5.6G VR II Lens (model: 1546) was used to image the flow every 30 seconds in large, fine, JPEG format, with 6000 × 4000 pixels per image. The camera was operated manually to ensure that the highest intensity within the images was less than the maximum of 255. This was also done for the calibration curve images. Upon capture, the images were imported into Matlab for
quantitative analysis. The raw red-green-blue (RGB) formatted images were converted to a hue-saturation-intensity (HSI) format. The intensity values were then used to determine the dynamic finger geometry and the vertical and horizontal water saturation profiles via Matlab. A statistical analysis was done for the parameters of the flow experiments utilizing a t-test using JMP Pro 12 from SAS.

**3. Results**

**3.1 Surface and Interface Characterizations**

The dynamic contact angles of each constituent solution at their various concentrations are displayed in Figure 5, with the stable contact angle measurements listed in Table 3 (stable measurements taken at 90 seconds). For most concentrations of the exudate solutions the contact



angles are significantly lower than the NaCl+HNS control solution (64.5 θ at 90 s), the exception being the low concentration of SRNOM. Statistical analysis of the contact angle and surface tension was done via t-test, and the differences are compiled in Table 4. Each constituent group displays a significant percent reduction in the contact angles of the experimental solutions

against the control at time 90 s, this significant difference is illustrated through the t-test groupings displayed in Figure 6. The SRNOM percent reduction in the contact angle values were 4% for 0.1 mg/L and 30% for 10 mg/L. The citrate percent reduction in the contact angle values were 18% for 0.1 mg/L and 29% for 500 mg/L. The oxalate percent reduction in the contact angle values were 15% for 0.1 mg/L and 26% for 500 mg/L. For the for tannic acid, however, the

variance in the reduction in contact angle was regardless of the increase in concentration, thus averaging the percent reduction is to 17%. Significant differences can be seen in all experimental solutions except for the low concentration of SRNOM. For the remained of the experimental solutions, there are two main groups that are not dependent on concentration. The first group that are similar to each other are tannic acid at both concentrations, citrate, and oxalate, both at their

low concentrations. Secondly, citrate, oxalate, and SRNOM at their high concentrations are similar to each other. However, the low concentration of tannic acid and the high concentration of oxalate, are not significantly different from each other.

Dynamic surface tension measurements were taken at 30 second intervals for all experimental

solutions and are displayed in Figure 7. Additionally, the stable surface tension measurements (at time 90 s) are listed in Table 3, and were used to determine the percent reduction in surface tension (Figure 8). All solutions exhibited a significantly lower surface tension than the NaCl+HNS control solution (75.7 mN/m at 90 s). As with the contact angle data, the changes in surface tension were not coupled directly with changes in concentration. For example, in Figures

5 and 7, the 500 mg/L citrate solution exhibited the lowest contact angle (46 θ) and the highest surface tension (73.6 mN/m), respectively (stable time at 90 s). The percent reduction in surface tension t-test groupings illustrate more specifically that the effect of the concentration variations on the surface tension is not consistent. The statistical analysis revealed two distinct groups; one consisting of the low concentration of oxalate and tannic acid and the high concentration of

citrate and SRNOM; and the other group consisting of the remainder, seen in Figure 8. It is important to note that, the percent reduction in surface tension for the experimental solutions

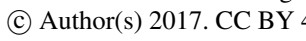



compared to the control were small, less than 10%. Additionally, the bond number for the experimental solutions was calculated to fall between 2.7-3.9E-4, illustrating the importance of the surface tension plays in the behavior of the solutions (Table 5). The densities of the experimental solutions were also calculated and found to range minimally, from 995.5 – 1002.8

$kg/m^3$.

## 3.2 Time Dependent Unstable Fingering Flow

The unsaturated gravity driven flow experiments were analyzed based on i) the number of fingers formed, ii) the velocity of finger propagation, and iii) the vertical and horizontal water saturation profiles. The two extreme concentrations for the plant constituents, which was 0.1

mg/L the lowest and 500 mg/L for the highest were compared with the concentration of the soil constituent, which was 0.1 mg/L to 10 mg/L.

In this comparison, the number of fingers formed in each experimental test solutions varied slightly (1.5 - 2.5 on average) with no discernable pattern between the various constituents and

concentrations. Although a variation was observed in the wetting depth of each solution, a relationship to the finger propagation velocity was observed in the citrate, oxalate, and tannic acid solutions. The relationship showed that the slower propagation of the fingers indicated a greater wetting depth (Table 3), which did not hold for the soil constituent SRNOM solutions. The finger length and the wetting depth of the citrate solutions at ten minutes is 22.4 cm and 3.5

cm for the low concentration, and 19.5 and 2.5 cm for the high concentration. The finger length and wetting depth of the SRNOM solutions at ten minutes is 17.7 cm and 7.0 cm for the low concentration, and 13.3 and 3.3 cm for the high concentration. The finger length and wetting depth of the oxalate solutions at ten minutes is 20.2 cm and 3.4 cm for the low concentration and 22.2 and 2.3 cm for the high concentration. The finger length and wetting depth of the tannic

acid solutions at ten minutes is 19.1 cm and 4.3 cm for the low concentration and 25.5 and 2.5 cm for the high concentration.

### 3.2.1 Finger Analysis

Matlab was next used to process and analyze the vertical water saturation profiles of the main finger for each solution (Figures 9 & 10). Although the NaCl+HNS control solution exhibited a



substantial wetting front and a uniform water distribution through the vertical finger profile, a steady gradient of higher to lower saturation, from the tip of the finger (~90%) to the top of the tank (~50%) was observed. The citrate solutions were similar in that both the 0.1 mg/L and 500 mg/L formed two fingers. However, the 0.1 mg/L citrate solution formed two long and narrow

fingers, width of 6.6 cm, with the majority of the water content located directly behind the wetting front (~100%). In juxtaposition, the 500 mg/L citrate solution formed two well-dispersed fingers that were 7.9 cm in width with relatively uniform water saturation profiles (~80%). These well-dispersed water saturation profiles characterized the oxalate solutions, with an average saturation of ~85% for the low and high concentrations. The 0.1 mg/L had a greater finger

length, most likely because the two fingers formed in the 0.1 mg/L sample had a width of 8.4 cm, unlike the three fingers with a width of 7.1 cm in the 500 mg/L analysis. Both long and narrow fingers with widths of 7.6 cm were observed forming from the tannic acid 500 mg/L test, with the majority of the water saturation occurring behind the tip of the wetting front (~100%). Further, the tannic acid at 0.1 mg/L had a slightly more uniform water saturation distribution

across the finger, with a width of 10.2 cm, though the majority of the saturation still occurred within the fingertip (~75%). Finally, SRNOM exhibited a deep wetting front and uniform water distribution across the finger profiles for both concentrations with only a minimal variation (~85-88%), though the low-to-high concentrations of the finger widths did vary from 6 cm and 7.5 cm respectively.

The horizontal finger saturation profiles were taken at 25%, 50%, 75%, and 90% of the main fingers length at 10 minutes (Figures 11 & 12). The greatest water saturation occurred within the middle finger of the NaCl+HNS control water saturation profile (~85%), which gradually dispersed towards the finger edges (~25-38%). Although both oxalate concentrations exhibited

similar saturations in the center of the fingers (~87%), the higher concentration had a greater dispersion zone (~40%) than the lower (~60%). A high level of saturation was observed in the substantial portion of the center of the main finger in the 0.1 mg/L citrate sample (~100%), with a small sharp zone of dispersion (~60%) along the edges of the finger. However, although less saturation (~85%) was observed in the middle of the finger of the 500 mg/L citrate sample, a

greater dispersion zone (~40-50%) did manifest along the finger's edge. SRNOM at both concentrations exhibited a similarly high saturation (~88%) in the center of the finger, though the





10 mg/L sample had a smaller zone of dispersion saturation (~50%). Overall, unlike the lower exudate concentrations that exhibited smaller water saturation dispersion zones toward the finger edges, at the higher concentrations, an increase in water saturation was observed through the center of the finger and larger water saturation dispersion zones were observed moving toward

the finger edges. Tannic acid responded differently than any of the other constituents, exhibiting lower saturation (~85%) in the middle of the finger and a greater dispersion zone (~60%) at 0.1 mg/L. At 500 mg/L, a higher saturation (~100%) in the middle of the finger and a smaller dispersion zone (~75%) were observed.

## 4. Discussion

The objective of this study was to elucidate the influence of various plant exudates and soil constituents on the soil infiltration, particularly wetting front instability and fingered flow in unsaturated porous media. By utilizing plant exudate and soil constituent solutions the physical effects of plant root interaction with soil infiltration processes and porous media spatial characteristics, such as heterogeneity and structure, were removed. Only the interactions between

the water chemistry, the porous media properties, and the flow dynamics were analyzed for an independent determination of the individual constituent solution effects on the soil infiltration processes that were in the form of wetting front instability and fingering flow phenomena.

Unstable fingering flow geometries have been studied under several scenarios, most notably that

of Parlange and Hill (1976) who derived modeling equations to predict the width of a finger in an air-water system by incorporating the sorptivity, and Glass and Nicholl (1996) who adjusted the equation to fit an air-water system. This system is expressed as:

$$d_c = \pi \frac{S^2}{2K_s(\theta_s - \theta_i)}\left(\frac{1}{1 - q_s/K_s}\right) \tag{2}$$

where $d_c$ is the minimum finger width, S is the sorptivity, $\theta_s$ is the saturated water content, $\theta_i$ is

the initial water content, $q_s$ is the flux, $K_s$ is the saturated hydraulic conductivity, and $\pi$ is a constant for a 2D system (Glass et al., 1991; Wallach et al., 2013). It is important to note that the intrinsic sorptivity is independent of the liquid properties (Philip, 1969; Wallach et al., 2013).

### 4.1. Infiltration Processes



The abrupt difference in water saturation along the fingers boundary, as observed in the vertical and horizontal water saturation distribution profiles that displayed high water saturation behind the wetting front and dry porous media ahead of the wetting front (Figures 9-12), can be explained by a reduction in sorptivity of porous media and therefore involving the contact angle

(Philip, 1971). The interplay between the resistance to wetting resulting from contact angle at the wetting front together with saturation overshoot and wetting front propagation is visualized by the vertical and horizontal water saturation distribution profiles that demonstrated in most cases the fingered flow with saturation overshoot (Figures 9 & 10) and decrease of contact angle (Figures 5 & 6) of all the system tested compared to the control infiltration experiment.

In the initial phase of infiltration process in dry porous media with contact angle close to zero, the wetting front diffuses though the porous media as the capillary forces dominate the gravity forces. Afterwards, as the capillary forces and the gravity forces assume an equal status, the wetting front propagates downward as a traveling wave. As the capillary force decreases from

the influence of the contact angle that either substantially differs from zero and/or is lower than the contact angle of the control solution, gravity forces overcome capillary forces. An abrupt wetting front is then generated with a high level of water saturation, which causes the development of an unstable wetting front that causes the formation of fingered flow. A higher (advancing) contact angle is located at the wetting front that propagates in the porous media and

a lower (receding) contact angle is present in the now-wetted porous media (Morrow, 1976). The disproportion between the inward and outward fluxes at the fingertip increases, which results in the development of the overshoot saturation. This increase in water saturation at the fingertip, "saturation overshoot" causes a rise in pressure and creates a positive matric potential gradient on top of the fingertip (DiCarlo et al., 1999; Geiger and Durnford, 2000). This rise in pressure

also increases the delineation between the wet and dry porous media at the edge of the wetting front and facilitates the propagation of the wetting front. The length of the saturation overshoot (i.e. the fingertip length ) was deemed a function of the wetting front velocities (DiCarlo, 2004; Wallach et al., 2013). Our experimental results which include the vertical water distribution profiles, the visualization of the overshoot saturation, and the wetting front velocity

measurements definitely support these findings (Figures 9 & 10; Table 3).



## 4.2. Redistribution Processes

After the initial infiltration process the redistribution within the fingering flow changes depending upon the simulated constituents and their concentrations. A noticeable effect of the constituents and concentrations upon the water distribution was observed, due to the different matric potentials within the systems. Although this increase in the matric potentials causes "saturation overshoot" (i.e. resulting in saturated fingertips), a reduced matric potential enhances the fluid movement in the porous media with a more uniform saturation distribution (DiCarlo, 2004; Glass et al., 1989c, 1989d; Wallach et al., 2013). The occurrence "saturation overshoot" at the fingertips occurs as the result of the fingering flow failing to overcome the pressure potential to ensure the downward flow (as seen in figures 9-12), most notably in the citrate and tannic acid flow systems. It is possible, however, to use data from the fundamental contact angle and surface tension analysis of the solutions in conjunction with the "saturation overshoot" observations to further illustrate the wettability changes in the flow patterns and the water content distribution of the finger saturation profiles. An increase in wettability causes an increase in solution sorptivity, caused by the reduction in the matric potential required for the solution to enter the pore spaces. These pressure changes present themselves as variations in the water saturation distributions determined elsewhere. Specifically, Wallace et al. (2013) found that for systems with higher contact angles, the air-entry pressure was greater, and thus the fingering flow is most likely to develop a more highly saturated fingertip (DiCarlo, 2004; Glass et al., 1989c, 1989d; Wallach et al., 2013). Similar results were determined in this present analysis in which the higher concentration solutions exhibited lower contact angles with more uniform saturation distributions. A comparison between the low and high concentrations for citrate and tannic acid clearly indicated the presence of "saturation overshoot" at lower concentrations when the contact angle was prevalent. At higher concentrations, however, the solution saturation exhibited a greater uniformity with a more substantially reduced contact angle. However, these conclusions cannot be applied to oxalate and SRNOM, which at both concentrations exhibit uniform saturations, illustrating that factors such as flow rate (Wallach et al., 2013), clearly affect the finger geometries and development. Additionally, with limited comparable plant exudate studies, comparisons can be drawn to existing studies upon the effects of surfactants in soil systems since plant exudates act as organic surfactants in the rhizosphere. It was determined that the simulated





plant exudates and the soil constituent solutions exhibited behaviors similar to that delineated in the studies of Bashir et al. (2011), and Henry and Smith (2002).

## 4.3. Exudate Effects

The addition of surfactants, biosurfactants, or exudates can alter the solution chemistry enough to produce a change in behavior of a media's hydraulic properties. In their study of hydraulic conductivity and hydrophobicity in root-soil interfaces, Zarebanadkouki et al. (2016) analyzed how wetting and drying cycling affected the hydraulic conductivity and potential hydrophobicity of the soil matric in the rhizosphere (Zarebanadkouki et al., 2016). They concluded that mucilage and exudates were a crucial determinant in conserving water for a plant to use during a drying period. However, after the water stores were depleted, the dried mucilage turned the soil hydrophobic, and remained so even after multiple re-wetting events. Furthermore the hydrophobicity caused a simultaneous reduction of the hydraulic conductivity of the systems (Zarebanadkouki et al., 2016). Juxtaposing these results that presented in this paper, the exudates, apart from the mucilage component, ultimately exert the opposite impact on flow through the porous media. This increase in the mobility of the fluid through the system is illustrated in Figures 9-12. Specifically, in their study of the effects of plant mucilage and phospholipids on the physical and chemical properties of soil, Read et al. (2003) observed a decrease of the surface tension of the soil water systems brought about by very small concentrations of phospholipid surfactants and exudate organic acids (Read et al., 2003). The increased wettability, and thus, mobility of the soil solution in the porous media is important for further studies to elucidate the transport of contaminants in the vadose zone and rhizosphere.

The rhizosphere is extremely complex, and interactions between the components are typically compounding, as a comparison of the study of Zarebanadkouki et al. (2016) to that of Walker et al. (2003) indicates. Specifically, in the broad approach used to elucidate the entire root system, Walker et al. (2003) found that the root exudates compounded with mucilage under certain circumstances exhibited a stabilizing effect on the rhizosphere. While the exudates lowered the surface tension of the soil pore-water, similar to what the authors determined in this research (Figures 7 and 8), the mucilage acted as a glue, bonding soil aggregates to the root sheaths and creating a water holding space around the roots as the surrounding soil continued to dry (Walker



et al., 2003). The simplicity of solely analyzing the effects of exudates as shown in Figures 5-8, is beneficial in understanding the separate components of the rhizosphere as they exist (i.e. uncompounded by the rest of the multiple components).

### 4.4. Imaging Capabilities

The recent surge in the development of visualization and modeling capabilities, specifically relating to 2D tanks, has been instrumental in studies attempting to elucidate rhizosphere systems and root zone development and characterization in situ (Carminati et al., 2016). These new capabilities, paired with current biological understandings have yielded a novel understanding of the entire plant-root-soil-water system. However, even with such an acceleration of research, the specific distribution and paths that the fluid takes from the soil matrix into and through the plant itself remains unknown (Carminati et al., 2016). As stated by Carminati et al. (2016), "*Coupling these measurements with imaging techniques that allow visualization of the root architecture and the structure and moisture of the rhizosphere could be the ultimate experiment to measure the rhizosphere properties*". The analysis presented here clearly indicates the effectiveness of combining 2D tank systems with imaging techniques such as LTM to furthering understanding of high special and temporal fluid distribution and moisture gradients in the rhizosphere directly under the influence of plant roots. The increase in sensor sensitivity, particularly of matric potential sensors or pressure transducers, will be of great value in acquiring additional insight to the mechanisms occurring in the rhizosphere.

### 5. Conclusion

Dynamic 2D tank flow experiments were coupled with the unique LTM in order to provide a novel analysis of the influence of plant exudate on infiltration flow and wettability in a porous media. This unique approach makes it possible for the rapid and high-resolution quantification of fluid saturation profiles. The primary objective of this research was to investigate the impact of low and high concentrations of simulated plant root exudates and soil constituents on unstable infiltration flows within a homogenous porous media. Overall, both concentrations of each constituent are observed to exert a significant effect on the wettability of the fluids when compared to the control. No changes were seen in between the different concentrations or





constituents in the flow systems. However, a change was seen between the experimental solutions as a whole and the control. Such findings illustrate the point that more normalized, or natural, concentrations of these constituents can significantly impact the wettability of infiltration into the rhizosphere. The data produced here provides the basis for future analysis of the

rhizosphere processes, where the natural biochemical concentrations exuded from the roots should be great enough to create significant changes in the mobility and wettability of fluids and anything moving within the root zone. The results developed here will also be of use in increasing the efficiency of agricultural practices related to irrigation application and water management.

**Acknowledgements**

This material is based upon work supported by the U.S. Department of Energy Office of Science, Office of Basic Energy Sciences and Office of Biological and Environmental Research under Award Number DE-SC-00012530.

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




**Table 1. Composition of Hoagland Nutrient Solution, recipe followed from Arnon and Hoagland (1940).**

| | Amount in volume (mL) | Chemical | |
|---|---|---|---|
| Main Solution (creates 1 L of solution) | 10 | 0.5 M $Ca(NO_3)_2$ | Calcium nitrate tetrahydrate |
| | 10 | 0.5 M $KNO_3$ | Potassium nitrate |
| | 4 | 0.5 M $MgSO_4$ | Magnesium sulfate |
| | 2 | 0.5 M $KH_2PO_4$ | Potassium dihydrogen phosphate |
| | 2 | Micronutrient Solution | |
| | Amount in mass (g) | Chemical | |
| Micronutrient solution (creates 500mL of solution) | 1.43 | $H_3BO_3$ | Boric Acid |
| | 0.9 | $MnCl_2$ | Manganese(II) chloride |
| | 0.11 | $ZnSO_4$ | Zinc sulfate concentrate |
| | 0.04 | $CuSO_4$ | Copper (II) Sulfate |
| | 0.1 | $(NH_4)_6Mo_7O_{24}$ | Ammonium molybdate |
| | 0.28 | $FeSO_4$ | Iron(II) sulfate heptahydrate |
| | 0.3 | EDTA | Edetic acid |



**Table 2. Concentration variations of constituents used to create the solutions tested in this work. Red indicates high concentrations of the constituent, yellow indicates a moderately high amount, and green indicates a typical natural concentration.**

| Concentration | Citrate (mg/L) | Oxalate (mg/L) | SRNOM (mg/L) | Tannic Acid (mg/L) |
|---|---|---|---|---|
| High | 500 | 500 | - | 500 |
|  | - | - | 10 | - |
| Low | 0.1 | 0.1 | 0.1 | 0.1 |




**Table 3. Compiled average experimental data from the triplicate fundamental interfacial characterization and the duplicate flow experiments.**

| Solution (mg/L) | Average Contact Angle (θ) | Average Surface Tension (mN/m) | Time (sec) | Average Number of Fingers | Average Velocity cm/sec | Average Wetting Depth (cm) | Average Length (cm) | Average Finger Width (cm) |
|---|---|---|---|---|---|---|---|---|
| **Fully Formed Finger Averages** | | | | | | | | |
| NaCl+HNS | 64.5 ±2.6 | 75.8 ±0.5 | 945 | 4 | 0.03 | 8.1 | 28.8 | 9.2 |
| Citrate 500 | 46.0 ±4.9 | 73.6 ±1.1 | 840 | 2 | 0.04 | 3.9 | 29.5 | 7.9 |
| Citrate 0.1 | 52.6 ±2.3 | 70.8 ±0.4 | 750 | 1.5 | 0.04 | 3.5 | 29.5 | 6.6 |
| Oxalate 500 | 47.3 ±2.3 | 71.6 ±0.6 | 720 | 2.5 | 0.04 | 2.3 | 27.5 | 7.1 |
| Oxalate 0.1 | 54.8 ±0.6 | 73.1 ±0.4 | 825 | 2 | 0.03 | 3.4 | 27.9 | 8.4 |
| Tannic Acid 500 | 54.4 ±2.7 | 70.8 ±0.3 | 660 | 1.5 | 0.05 | 2.5 | 29.9 | 7.6 |
| Tannic Acid 0.1 | 52.3 ±1.2 | 72.7 ±0.4 | 930 | 2 | 0.03 | 5.5 | 27.8 | 10.2 |
| Organics 10 | 45.0 ±2.6 | 73.4 ±0.4 | 975 | 2 | 0.03 | 5.0 | 29.2 | 6.0 |
| Organics 0.1 | 61.7 ±2.6 | 71.1 ±0.2 | 870 | 2.5 | 0.03 | 9.8 | 25.7 | 7.5 |
| **10 minute Finger Averages** | | | | | | | | |
| NaCl+HNS | | | 600 | 4 | 0.03 | 6.1 | 17.3 | 7.7 |
| Citrate 500 | | | 600 | 2 | 0.03 | 2.5 | 19.5 | 7.2 |
| Citrate 0.1 | | | 600 | 1.5 | 0.04 | 3.5 | 22.4 | 6.1 |
| Oxalate 500 | | | 600 | 2.5 | 0.04 | 2.3 | 22.2 | 7.0 |
| Oxalate 0.1 | | | 600 | 2.5 | 0.03 | 3.4 | 20.2 | 7.4 |
| Tannic Acid 500 | | | 600 | 1.5 | 0.05 | 2.5 | 25.5 | 7.5 |
| Tannic Acid 0.1 | | | 600 | 2 | 0.03 | 4.3 | 19.1 | 7.8 |
| Organics 10 | | | 600 | 2 | 0.02 | 3.3 | 13.3 | 6.0 |
| Organics 0.1 | | | 600 | 2.5 | 0.03 | 7.0 | 17.7 | 6.6 |





**Table 4. Statistical analysis summary table for t-test significant differentiations.**

| Solution | Contact Angle (@90s) | Surface Tension (@90s) | # Fingers (@10min) | Velocity (cm/sec) (@10min) | Wetting Depth (cm) (@10min) | Length (cm) (@10min) | Avg Width (cm) (@10min) | # Fingers (@full) | Velocity (cm/sec)(@full) | Wetting Depth (cm) (@full) | Length (cm) (@full) | Avg Width (cm) (@full) |
|---|---|---|---|---|---|---|---|---|---|---|---|---|
| Control | A | A | A | AB | AB | AB | A | A | B | AB | A | AB |
| CitrateHigh | D | B | B | AB | AB | AB | A | B | AB | BC | A | C |
| CitrateLow | BC | C | B | AB | AB | AB | A | B | AB | BC | A | ABC |
| OxalateHigh | CD | C | B | AB | AB | AB | A | B | AB | BC | AB | ABC |
| OxalateLow | B | B | C | AB | B | AB | A | B | AB | C | AB | BC |
| TannicHigh | B | C | B | AB | AB | AB | A | B | B | ABC | AB | A |
| TannicLow | BC | B | B | A | A | A | A | B | A | BC | A | BC |
| OrganicHigh | D | B | B | AB | A | AB | A | B | B | A | A | BC |
| OrganicLow | A | C | B | B | AB | B | A | B | B | ABC | B | C |





**Table 5. Experimental solution densities and bond numbers. Reported numbers are the averages of triplicates.**

| Solution | Average Density (kg/m³) | Bond Number |
|---|---|---|
| Control | 997.6 | 2.90E-04 |
| Citrate High | 996.8 | 2.99E-04 |
| Citrate Low | 999.3 | 3.11E-04 |
| Oxalate High | 1002.8 | 3.09E-04 |
| Oxalate Low | 997.2 | 3.01E-04 |
| Tannic Acid High | 1000.2 | 3.11E-04 |
| Tannic Acid Low | 998.5 | 3.03E-04 |
| SRNOM High | 998.3 | 3.00E-04 |
| SRNOM Low | 995.5 | 3.09E-04 |


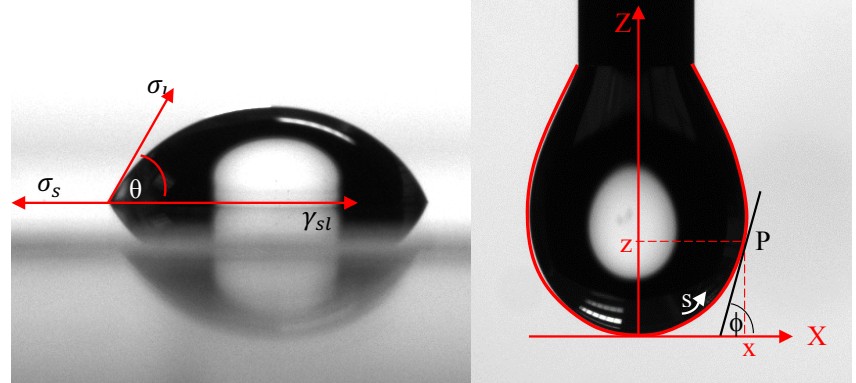

**Figure 1. Visual of the drop shape analysis of (Left) contact angle and (Right) the surface tension pendant drop using the Kruss Easy Drop. The images are of the control solution (HNS+HNS) taken at 90 seconds. Figure is based off of figures appearing in the Kruss User's Manual (Kruss, 2010).**

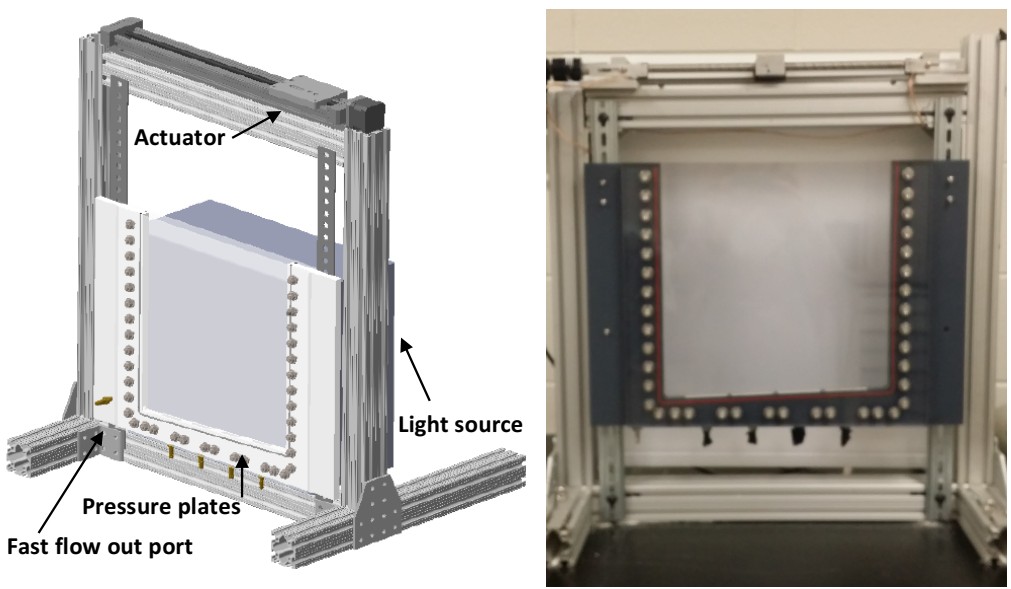

**Figure 2. Experimental set up of the 2D tank system.**





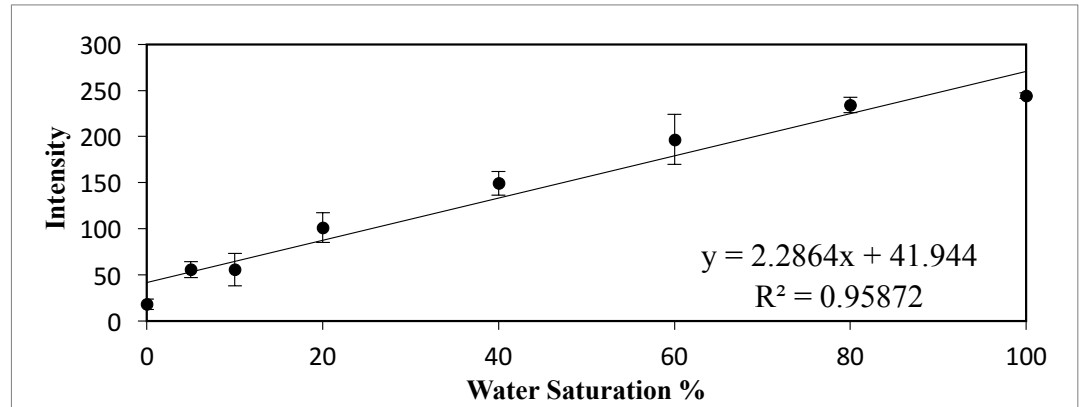

**Figure 3. Empirical calibration curve of the transmitted intensity vs. the water saturation, with its linear trend line and slope equation.**

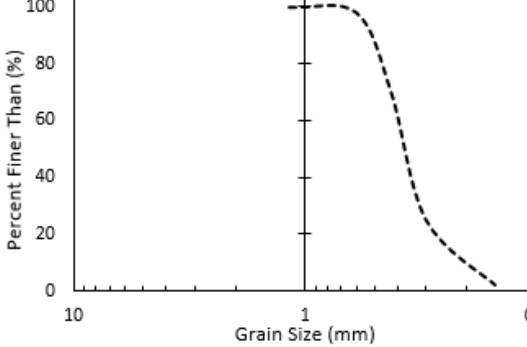

| USA STND Sieve Mesh Size | Grain size (mm) | % of sand passing through (smaller than mm size) |
|---|---|---|
| 16 | 1.18 | 100 |
| 30 | 0.6 | 98 |
| 40 | 0.425 | 70 |
| 50 | 0.3 | 25 |
| 100 | 0.15 | 2 |
| pan (silts) | | 0 |

**Figure 4. Characteristics and properties of the US Silica C778 ASTM graded sand.**




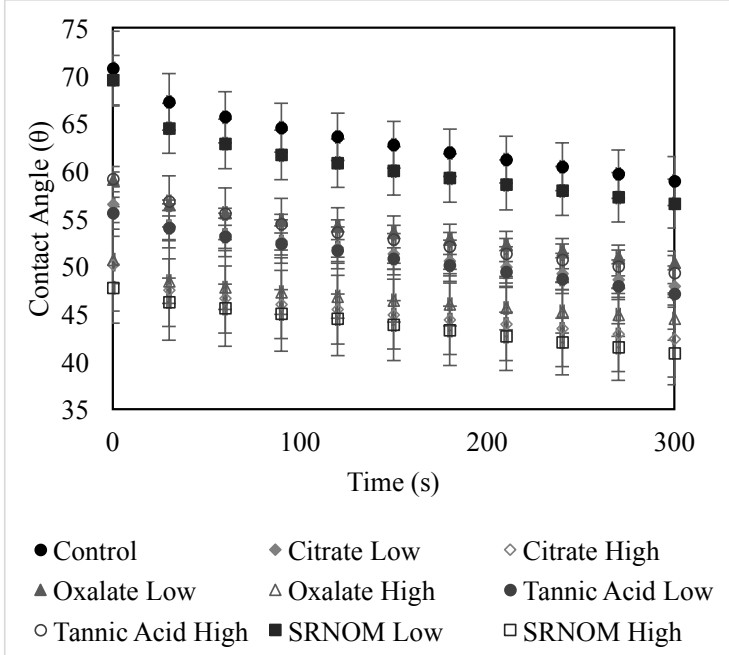

**Figure 5. Dynamic contact angles of experimental solutions. Averages graphed, and associated error bars represent one standard deviation.**




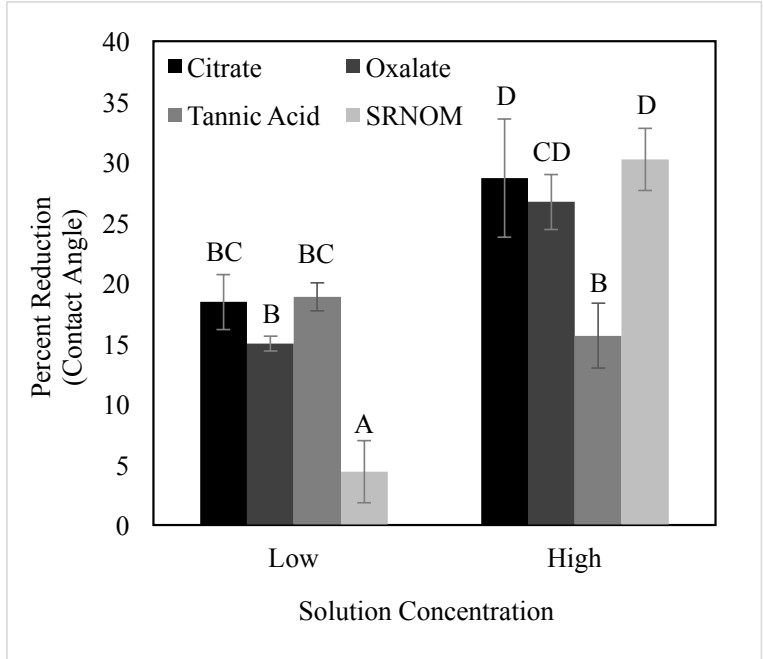

**Figure 6. Percent reduction in contact angle from the control (group A), changes at stable time (90 seconds). Averages graphed and associated error bars represent one standard deviation.**





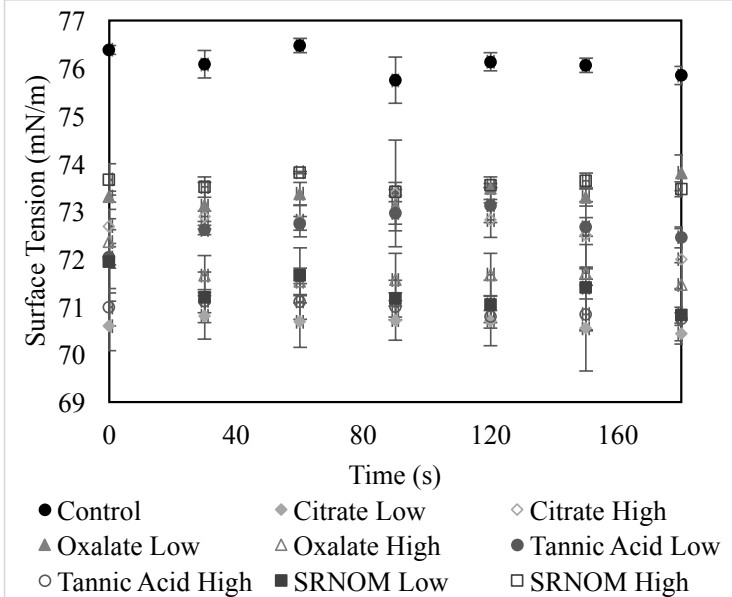

**Figure 7. Dynamic surface tension of experimental solutions. Averages graphed, and associated error bars represent one standard deviation.**



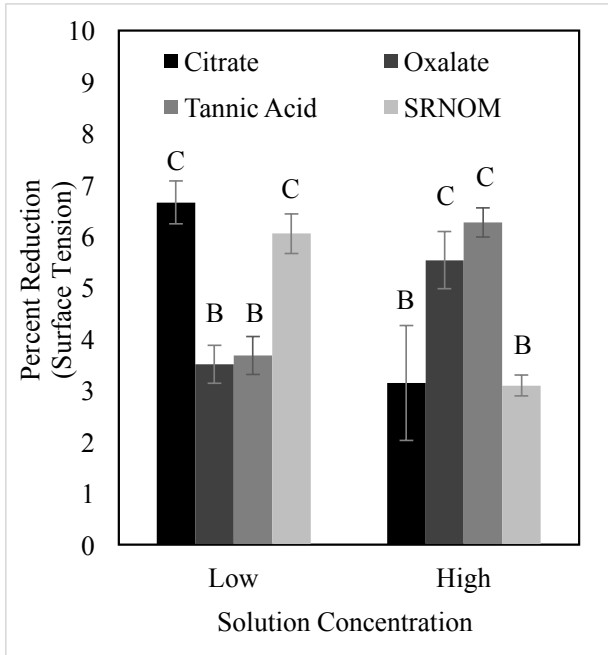

**Figure 8. Percent reduction in surface tension from the control (group A), changes at stable time (90 seconds). Averages graphed and associated error bars represent one standard deviation.**





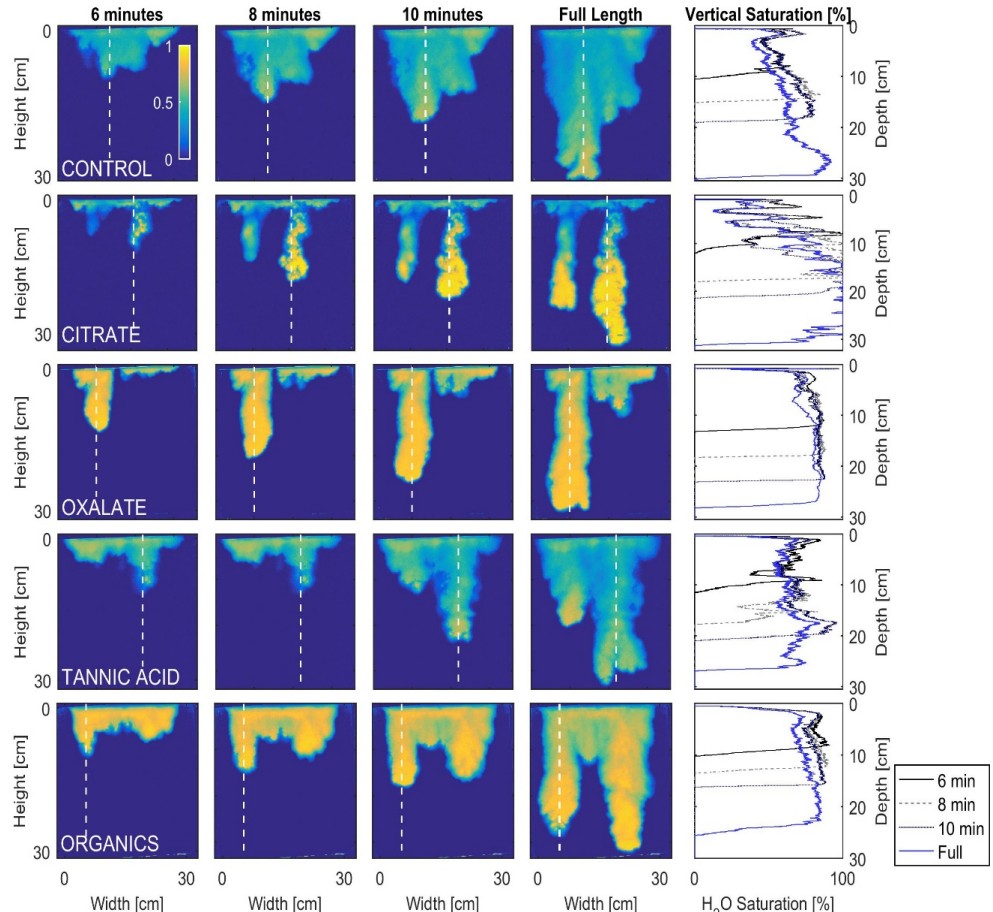

**Figure 9. Multigraph comparison of the dynamic vertical profiles for low concentration exudates. The images are of the main finger over time.**



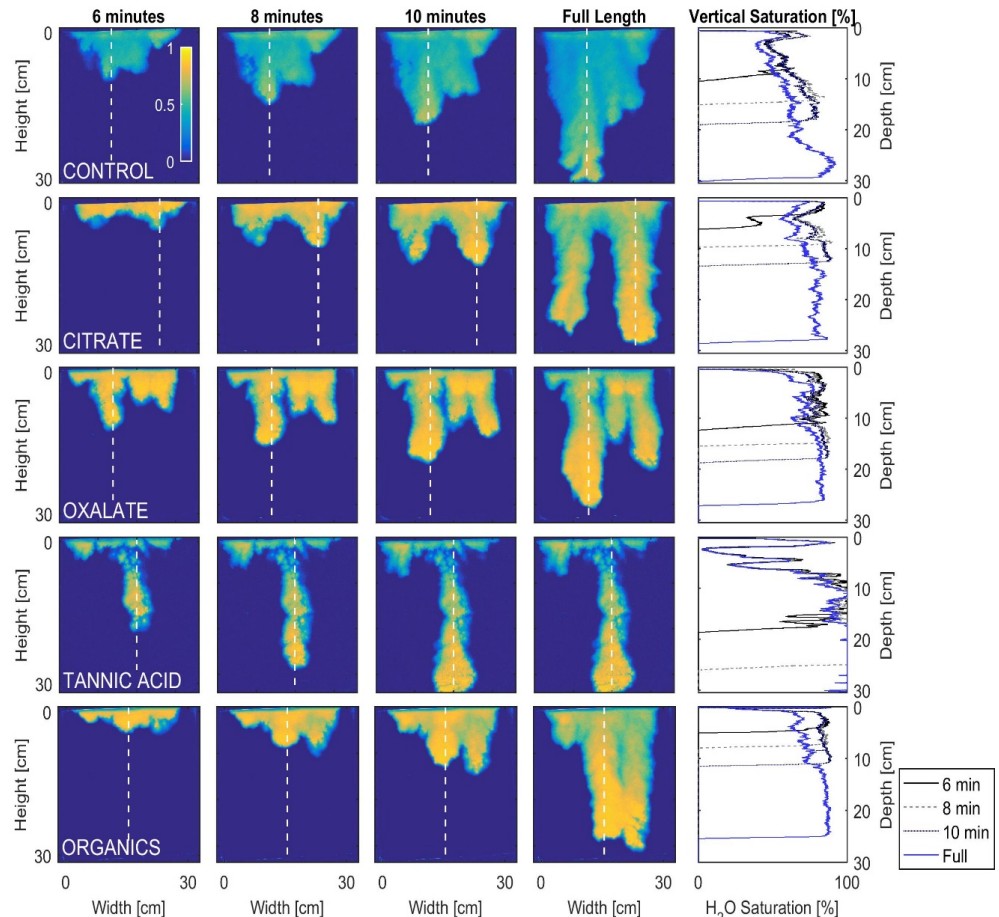

**Figure 10. Multigraph comparison of the dynamic vertical profiles for high concentration exudates. The images are of the main finger over time.**



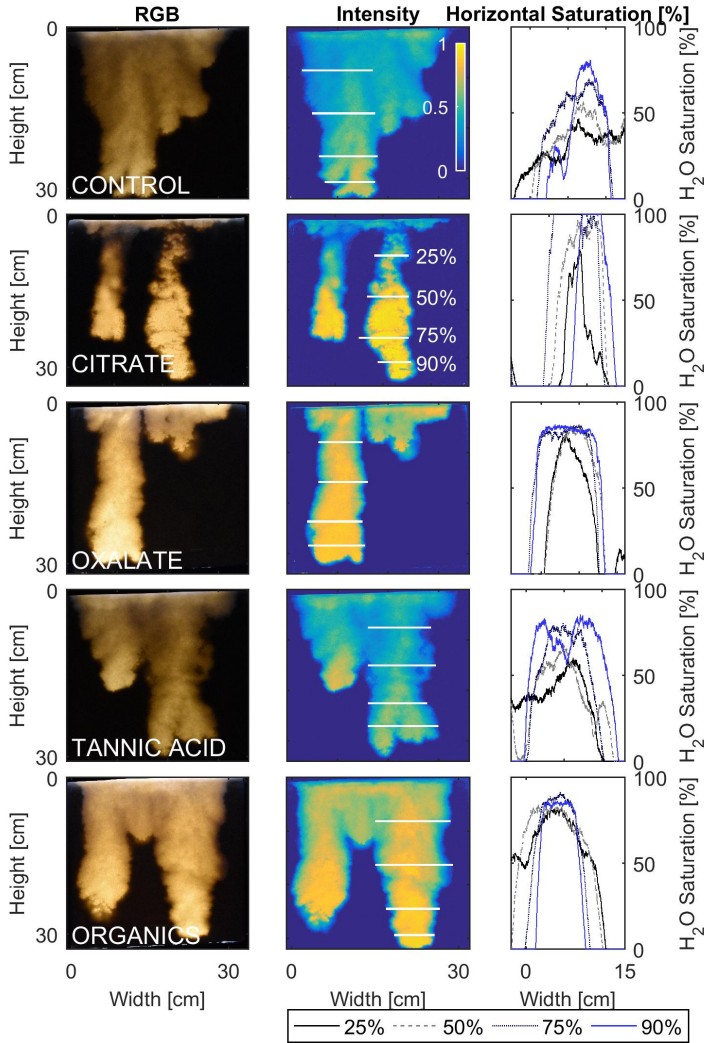

**Figure 11. Multigraph comparison of the width profiles for low concentration exudates. The images are of the main finger at its full length. The location of the widths are measured at 25%, 50%, 75%, and 90% of the total fingers length.**




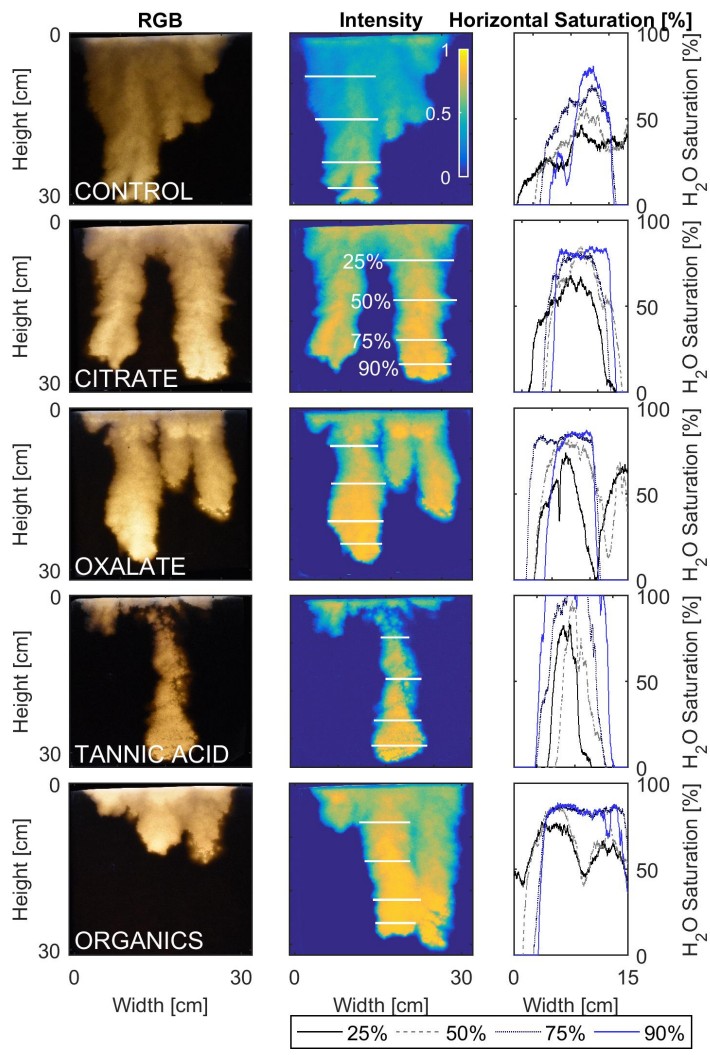

**Figure 12. Multigraph comparison of width profiles for high concentration exudates. The images are of the main finger at its full length. The location of the widths are measured at 25%, 50%, 75%, and 90% of the total fingers length.**



5                                     APPENDICES



Appendix A

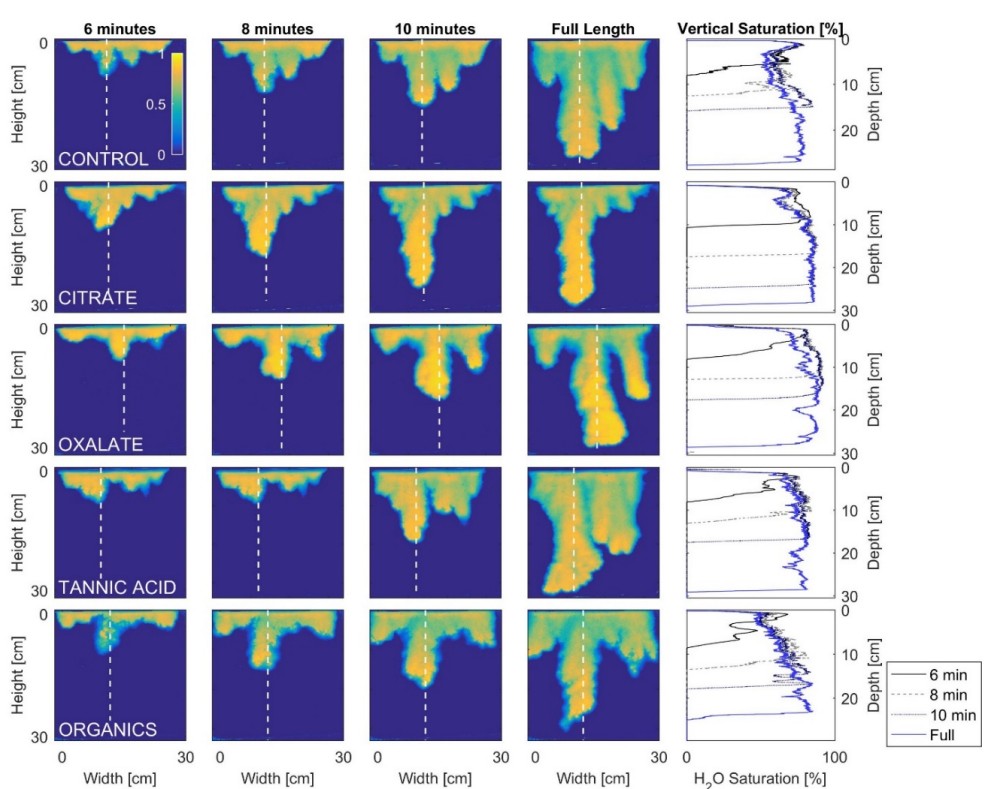

**Figure 1A. Second tank experiment data set. Multigraph comparison of the dynamic vertical profiles for low concentration exudates. The images are of the main finger over time.**





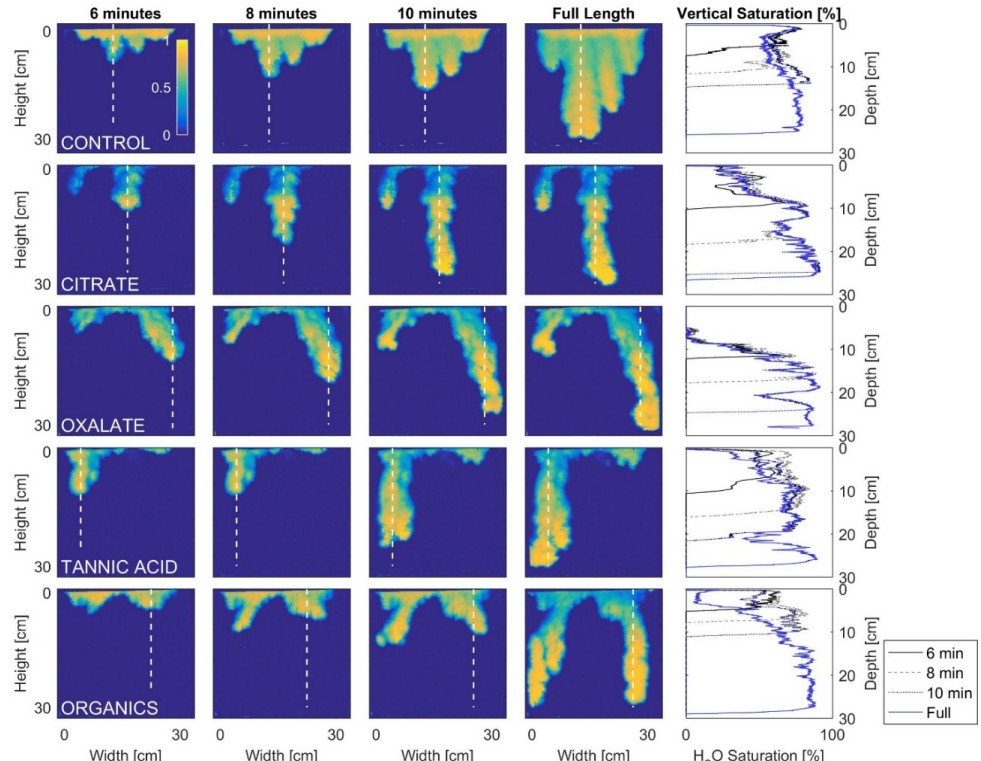

5 **Figure 2A. Second tank experiment data set. Multigraph comparison of the dynamic vertical profiles for high concentration exudates. The images are of the main finger over time.**



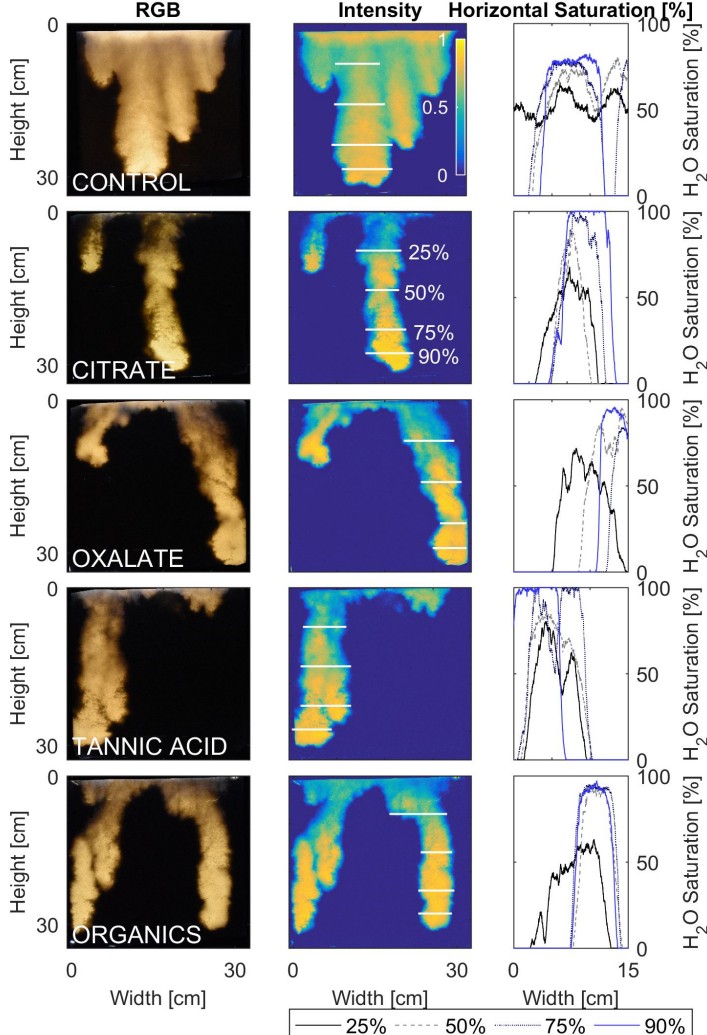

**Figure 3A. Second tank experiment data set. Multigraph comparison of width profiles for high concentration exudates. The images are of the main finger at its full length. The location of the widths are measured at 25%, 50%, 75%, and 90% of the total fingers length.**





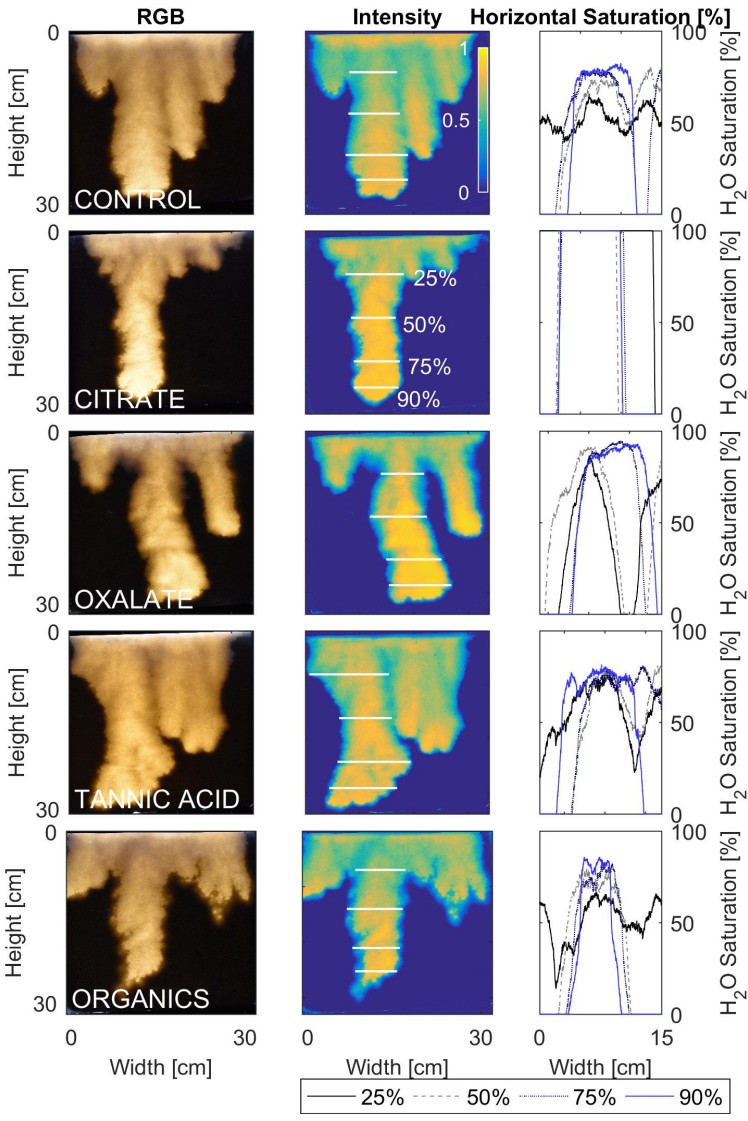

**Figure 4A. Second tank experiment data set. Multigraph comparison of width profiles for low concentration exudates. The images are of the main finger at its full length. The location of the widths are measured at 25%, 50%, 75%, and 90% of the total fingers length.**