# Peer review of "Preferential Flow Systems Amended with Biogeochemical Components: Imaging of a Two-Dimensional Study"

_Hydrology and Earth System Sciences, 2017_

## Short Comment (SC1) · 28 Aug 2017

General comments:

The work is very interesting, relevant, and contributes to our need for a more complete understanding of how the complex soil physics and biochemistry of the soil zone impacts vadose zone processes. The technical work appears to be very well performed and no concerns are raised. The comments below are all associated with the communication of the work and how it fits into present day understanding and existing literature. A few suggestions are made for additional calculations, figures and tables.

Overview comments

1. The paper could be shortened. The introduction and background is too extensive. Unless there is an intent of making this paper a review paper on preferential flow, the introduction should be limited to a brief note of the importance of preferential flow and then be focused on the specific background that is relevant to this study, which is the effect of fluid properties on finger flow development, stability, and the presence of different solutes in the rhizosphere.

2. The Bond number is calculated but its relevance is never discussed in the Background or Discussion sections.

3. There is too much emphasis on the statistics of the contact angle and surface tension. The number of figures and the discussion can be reduced. This work is about how these two parameters affect finger flow development, and does not an attempt to characterize the relationship between interfacial properties and concentration for different solutions. The information on contact angle, surface tension, fluid densities, Bond number, etc. can be captured in a table summarizing fluid properties (I realize that this is in separate tables at present), eliminating the need for figures 6 and 8. The graphs showing time dependent values of contact angle and surface tension are useful.

4. The discussion on finger geometry (or morphology), saturation overshoot, finger width and moisture content distribution needs some improvement. For example, (A) Finger properties are compared to contact angle values and solution concentrations, but a more complete analysis would be to compare finger geometry to capillary pressure at the wetting front. This may solve the problem that tannic acid (whose contact angle does not change with concentration) contradicts the apparent general result. (B) A graph of finger properties with capillary pressure (using the Young-Laplace equation) which takes both contact angle and surface tension into account would rapidly demonstrate in a visual fashion any trends seen.

5. The technical work, results and data are valuable but the discussion and conclusions

are weak. Some of the concepts associated with capillarity and sorptivity and miss-stated.

6.Equation 2 is noted in the Discussion section but not used in the work. Should this equation be moved to the background section? Can it be used to compare observations with predictions? 7. Significant editorial work is necessary before it is ready for publication or public review. A few editorial comments are noted below but the focus of this review is on the technical content.

SPECIFIC COMMENTS

Abstract

1. Line 27. Should 'processes' be 'properties'?

2. Line 28. Change to " …the control with a contact angle of 64.5 deg, and a surface tension of 75.75 N/m…" There are three problems with the sentence as is: (1) You have not defined the symbol 'theta' as a contact angle yet. (2) even if you had defined it, "64.5 " is not traditionally how it is written but rather it is written " = 64.5". Same changes required on line 29. (3) I thought surface tension units were N/m, not Nm/m.

3. Line 29: "changes" should be "differences". The last sentence is not a complete sentence. Page 2

4. Line 7: change 'element of' to 'contributes to'

5. Line 9: change 'especially' to 'by controlling" and add 'supporting prior to vegetation.

6. Line 10: sentence starting with 'consequently', needs to be move to the end of this paragraph. As is, 'consequently' is the incorrect word because the previous statement discusses the existence of water and life at the land atmosphere interface, not groundwater issues. You need to add the logical linkage, that in addition to controlling soil moisture and plant life, soil controls deep drainage of infiltrated water, its rate and quality.

1. Line 22: 'without' should be 'with'

2. Line 23: change 'instance' to 'interaction'

3. Line 25:

4. The soil hydraulic properties (aka, permeability) and the moisture content are properties that 'set the conditions but the instability is caused by the interplay between the two forces, capillarity and gravity.

5. Line 2: "thus" insinuates that the fact that there are two forces leads to instability. IN reality, both stable and unstable regimes are affected by both gravity and capillary forces. It is when the forces are of nearly equal but opposite in magnitude, that small irregularities, such as small heterogeneities (which are inevitable, even in a practically homogeneous media) will lead to the condition where gravity dominates, at that point the wetting front breakthrough will occur and a finger will begin to form. Re-reading the glass papers may clarify this point. All instability conditions require a trigger to progress into the next favorable stage.

6. The next two paragraphs (starting with 'The non-linearity', and 'In addition)are too extensive since this paper does not discuss how any of the work will affect or change the present understanding of the mathematical or numerical work. I think the authors could move directly from the points made in the first paragraph of page 4, to the fact that change in contact angle will control capillarity and therefore the propensity for instability. Then mention the relevant work associated with this issue, such as the repellency work, etc.

Page 5.

7. The paragraph starting with 'The vadose zone…" I do not think the vadose zone

needs to be defined here, again. The instability you are study8ing is occurring due to plant and soil solutes, so the soil zone is what is relevant.

8. The paragraph starting with 'The plant root. . .' is interesting by not relevant to this paper. The change in repellency caused by the dehydration of the muscilage is a change in the properties of the solid, not the solute. This paper focuses on the properties of solutes.

9. The paragraph starting with "Indeed" is not necessary. While noting good examples of serious groundwater contamination, these may not be relevant to development of unstable finger flow issues, nor are these associated with plant exudates. In addition, the two examples given, while associated with preferential flow, they are more closely related to fracture flow than unstable finger flow. More relevant to finger flow instability through soil media are agricultural contaminants.

10. Line 10: regarding the word monitored. The dynamics were not monitored via surface tension or contact angle. The dynamics were monitored via visualization. The fluid properties were quantified by measuring contact angle and surface tension. 11. Line 25: what is the relevance of a hydroponic system? 12. Line 25: why was NaCl+HNS rather than DI water used as a control? Not suggesting its wrong, but should explain the rationale.

13. Materials and methods: you state that statistics were performed on the contact angle and surface tension measurements, but there was no mention of how many reps where done.

14. Line 9-14: It is unclear to me how the calibration was performed. How did you measure the moisture content for the different intensities? Or did you preset the moisture content and then measure the intensities? Also, I did not understand the comment about the 80%.

15. Line 18: what does it mean that the 0% and 100% intensity values "were found"? Did you look for locations where you had 100% and 0% and measure the moisture content there?

16. Line 22: describe the packing device? Was the sand poured in? How did you avoid horizontal layering of sand during packing?

17. Line 14: the "flow velocity" is the velocity of what? The fluid in the peristaltic pump line, or the raindrop speed, or the surface area averaged input flux rate?

18. Line 29: dynamic contact angle: the term "dynamic" contact angle is usually associated with the dependence of the contact angle on interface velocity, or in a larger scale the wetting front velocity, but fig 5 it is graphed per unit time, so is this the time-dependent contact angle or the dynamic contact angle?

19. Line 9: delete the extra "for"

20. You report contact angle values through time up to 90 seconds, and discuss in detail how much these angles have changes over time. But, you do not discuss the significance of these changes to the dynamics of the finger flow system.

21. Line 19: is this "dynamic" or "time dependent"?

22. Line 22: add "over time" following the word 'tension'

23. Line 23 and 24: the word "changes" should be "differences"

24. Line 26: delete the word 'respectively'

25. Line 28: sentence needs clarifying that the 'groups' are of surface tension values.

26. Line 9-11: since none of your graphs or images use the terms "plant constituents" or "soil constituents" you should either include their proper names in parenthesis to help the reader.

27. Lines 13-26: can this be put into a table format?

28. Liens 1-19: It may help to illustrate the differences if the data was grouped according to contact angle, surface tension or Bond Number groups rather than chemistry, since it is those properties that should control finger formation.

29. Line 21-31: can this information be displayed in graphical form? The changes in finger saturation cross-section are interesting. Again, it would be interesting to see if there is a grouping according to chemistry concentration, or interfacial properties.

Page 16:

31. Line 27: you mention the intrinsic sorptivity but there is no linkage to the relevancy of this statement to the work presented here. It is true that the intrinsic sorpitivity is independent of fluid properties because it is a normalized property.

32. What is the relevancy of Equation 2. It was not used in the work? It would be valuable to see how the predictions based on this equation compare to the observations.

33. Top paragraph needs to be improved for clarity. The statement "reduction in sorptivity" and "decrease in contact angle" suggests that it was reduced from some earlier value. But in truth the reason there is overshoot is that the sorptivity at the front is lower

than the hydraulic conductivity of the media above it. At the wetting front the advancing contact angle is greater, and therefore the sum of capillary and gravitational forces results in lower rate of forward motion. The pressure overshoot is simply associated with hydraulic pressure of water building up above the (way too slow) wetting front. It is also very likely that what is being seen at the front is the 'dynamic contact angle" which is a contact angle that is even greater than the static contact angle measured at 90 sec mark. The dynamic effect is caused by a competition between inertia, motion below the interface, the flexibility of the interface and the movement (stickiness) of the triple line. The contact angle changes in value with the speed of the wetting front. While it may be valuable to measure the dynamic contact angle of these substances, it is not an easy measurement to take. However, you may be able to obtain it from data you already have since there are a few theoretical relationships that can approximate the value of the dynamic contact angle based on the Capillary Number, the static contact angle and the velocity. There is a relationship between the Capillary Number and the Bond Number.

34. Line 11: why would the contact angle be close to zero during the initial stages?

35. Lines14-16: This sentence is confusing. The way it's written, the first half appears to contradict the second half.

36. Line 1: use "differs" instead of "changes"

37. Line 2: change "simulated ….concentrations" to "solution chemistry and concentration."

38. Line 4-5: The sentence is unclear. Also, you note a relationship between finger morphology and matric potential of the media. Did you measure the matric potential?

39. First paragraph: There is quite a bit of repetition with two previous paragraphs where the overshoot mechanisms are also described.

40. Line 13: What is meant by "wettability changes in the flow patterns", do you mean how the wettability changes the flow pattern? Or that the wettability is changing within the flow pattern?

41. Line 16: what is meant by "present themselves as" do you mean "result in" ?

42. Line 21, add "(except for Tannic acid)" following the word 'angles'

43. Line 22: by 'prevalent' do you mean 'greater'?

44. Line 22-24: I think that the comparison needs to be made between finger geometry and capillary potential (using Young-Laplace eq), not individually with contact angle. The reason for this is that you will note that tannic acid did not show a difference in contact angle between low and high concentrations, but it did show a large difference in surface tension, and in finger geometry. The capillary pressure calculation would capture that difference. This applies through line 27 in this paragraph.

45. Line 1: Describe the behavior observed by Bashir etc., so that the reader does not have to go to that citation to understand what is being discussed.

46. Line 5: the media's hydraulic properties do not change, it is the fluid's properties that are changing.

47. Line 12: I do not follow how the research presented in this paper is relevant to a discussion associated with the apparent repellency effect of dry mucilage. The study is of a wetting experiment onto dry mineral base (not organics), with a well saturated organic in solution.

48. Line 13-21: Section needs restructuring. What is being said is unclear. Wrong words are being modified, and some of the thoughts are incomplete.

49. Paragraph starting on line 23 needs sharpening. (A) Line 23-25: sentence needs restructuring. (B) Line 26: mucilage is a type of root exudate. (C) Line 27: what is

meant by "stabilizing"? mucilage does act as a glue, and it does create interstitial spaces but it also absorbs water by osmotic processes.

50. Line 16: change 'special' to 'spatial'

51. Line 26-27: change "both...constituent" to "solution chemistry and concentration". Change "when compared to the control" to "and the resulting infiltration profile"

52. Line 28: No changes in what?

53. Line 2: What is meant by "more normalized"

54. Line 7: delete "or anything", this paper is focused on fluids.

55. Line 8: do you mean efficiency or effectiveness?

---

## Referee Comment (RC1) · Anonymous Referee #1 · 5 Oct 2017

This paper focuses on the influence of the soil rhizosphere's plant exudates on soil wettability and infiltration. Specifically, the authors study the spatial and temporal development of unstable flow leading to preferential flow "fingering" and their relationship with those plant exudates and soil solution components. The authors propose that these compounds are responsible for keeping the soil around the roots moist by swelling and adsorbing water, and also to keep the contact between the roots and soil particles. To study the effect on preferential flow, the authors employ a 2D tank filled with silica sand equipped with a rainfall simulator on the top of the apparatus in order to create the necessary flow conditions for the infiltration experiments. On the back of the experimental device, a light panel was place so the development of such "fingers" could be track by

the so-called light transmission method (LTM). They also measured the contact angle and surface tension of the solutions in the soil by using a Kruss easy drop SA1. To simulate plant exudates they selected sodium citrate and sodium oxalate, as soil components they used Suwanee River Natural Organic matter (SRNOM) and tannic acid, and as a base solution NaCl+ a nutrient solution. Through these series of experiments, they analyzed (i) the number of fingers formed, (ii) the velocity of their propagation, and (iii) the vertical and horizontal water saturation profiles. This rich data set is used to test main relationships among water chemistry, the porous media properties and the flow dynamics of each individual solution and its effect on the soil infiltration process.

The manuscript is well posed and written and of interest to HESS readers. It is novel as it addresses an important topic for which there is limited knowledge to date. The work is consequential; if plant roots promote the creation of unstable soil water flow, many soil remediation applications will need to be reassessed. Previous literature has theorized these effects but none (or at least none to this reviewer knowledge) studied them experimentally. For instance, the work shows how the addition of surfactant exudates into the soil can alter the solution chemistry enough to produce a change in behavior of the porous media's hydraulic properties, by exudates increase wettability and thus mobility of the soil solution in the porous media. I suggest minor revisions listed below.

**Specific Comments**

Ln 12: Possible typo: 'though' instead of 'through' Ln 16: Possible typo: 'special' instead of 'spatial'

Figures 5 and 6: Although the results are well presented in their corresponding tables, in the figures 5 and 6 it is very difficult to differentiate between the different components and their respective deviations.

---

## Referee Comment (RC2) · Anonymous Referee #2 · 17 Oct 2017

The manuscript presents experimental results of unstable flow patterns in sand boxes using a light transmission technique that allows identifying differences in flow patterns caused by the composition of the solution of the irrigation water. Plant exudates and soil solutions with different contact angles and surface tensions were tested and related to the effects on the flow finger development. Results demonstrate quite different patterns during the infiltration process. The test comparing various solution compositions is new and the experiments are highly sophisticated and carefully carried out, especially the combination with the light transmission method that allows determining local water contents is innovative. But the manuscript could be better structured, shortened, and more focussed on the analysis of these experiments.

Claiming this manuscript to be on original research, my immediate impression was that authors should come to the main points more quickly; many references are not further used for the idea and results of this study. When continuing reading, the review part appears more and more excessive; in particular, the multiple referencing is changing the appearance towards a review article in which authors are trying to collect all relevant papers. Such an overview of the literature is quite nice and could be the basis for a separate manuscript. And despite the large number, referencing is still limited, for example, P5 L4: "...fronts has been studied primarily in two-dimensional tanks...", recently also 3D patterns observed using geo-electrical imaging (e.g., Ganz et al., VZJ 2014, doi:10.2136/vzj2013.04.0074). Furthermore, the specific research hypotheses are not so explicitly stated in the introduction (more indirectly somehow within the review), so that the idea of the experiments and reasons for doing it as it was done remained unclear to me at the end of the introduction, where also the objectives were too general. Clear objectives statements are then found in the discussion and again in the conclusions. The methods are explained very detailed, for Tables and Figures, however, I found it very difficult to understand without having the abbreviations explained in headers and captions (e.g., Suwanee River Natural Organic Matter (SRNOM) acronym etc). One methodological problem that was probably discussed in earlier papers on the technique (?) was unclear to me. This is how to obtain repeatable uniformly compacted sand samples so that the packing effects are not influencing the effects of the solution composition. The stated accuracy of bulk density value (1.5043) with 4 digits is quite ambitious. Wetting and especially the partial wetting during the infiltration may change the arrangement of sand particles such that the pore structure may not be always constant. Although results of all three replicate infiltration experiments are provided, the question whether each of the finger pattern is characteristic for each solution and comparable for the replicate is not clear to me. I like the detailed explanation of results but data analysis seems still a bit limited. The hypotheses and how the results could be applied to soils remained unclear. Detailed comments 1. The abstract reads well, I only wondered if the conclusions here correspond to those in the
conclusion chapter. 2. Page 7, Lines 15-25: not necessary and unclear 3. Discussion: Starts with the objectives, first paragraph contains hypothesis and should appear as part of the introduction. I was wondering how was equation 2 used? 4. Page 17, Lines 22-30: This is more or less an introduction to the closer topic and the results seem to confirm existing knowledge. 5. Page 19, Lines about 5-11: This is doubling introduction 6. Conclusions chapter gives more a summary of results than conclusions.

---

## Editor Comment (EC1) · B. Berkowitz (Editor) · 18 Oct 2017

To the authors: Please note that the "SC1: Technical Review" by Maria I Dragila should be treated as a formal solicited review, and so you are also requested to reply to this comment with a detailed response letter, along with the other two formal reviews ("RC1, RC2"). Thank you.

---

## Author Comment (AC1) · 24 Nov 2017

To the authors: Please note that the "SC1: Technical Review" by Maria I Dragila should be treated as a formal solicited review, and so you are also requested to reply to this comment with a detailed response letter, along with the other two formal reviews ("RC1, RC2"). Thank you.

Response: The authors thank Editor Brian Berkowitz for his comments. We will be treating the "SC1: Technical Review" by Maria I. Dragila as a formal solicited review and provide a detailed response letter, along with our replies to the other two formal reviews ("RC1, RC2").

---

## Author Comment (AC3) · 24 Nov 2017

OVERVIEW COMMENTS

The manuscript presents experimental results of unstable flow patterns in sand boxes using a light transmission technique that allows identifying differences in flow patterns caused by the composition of the solution of the irrigation water. Plant exudates and soil solutions with different contact angles and surface tensions were tested and related to the effects on the flow finger development. Results demonstrate quite different patterns during the infiltration process. The test comparing various solution compositions is new and the experiments are highly sophisticated and carefully carried out, especially the combination with the light transmission method that allows determining local water contents is innovative. But the manuscript could be better structured, shortened, and more focused on the analysis of these experiments.

Response: The authors thank Referee #2 for reviewing our manuscript and for the comments about our research: "Results demonstrate quite different patterns during the infiltration process. The test comparing various solution compositions is new and the experiments are highly sophisticated and carefully carried out, especially the combination with the light transmission method that allows determining local water contents is innovative." We will follow the reviewer's suggestions for our manuscript to "be better structured, shortened, and more focused on the analysis of these experiments." We respond below to each of the reviewer's comments and provide detailed information about the proposed revisions.

GENERAL COMMENTS

Claiming this manuscript to be on original research, my immediate impression was that authors should come to the main points more quickly; many references are not further used for the idea and results of this study. When continuing reading, the review part appears more and more excessive; in particular, the multiple referencing is changing the appearance towards a review article in which authors are trying to collect all relevant papers. Such an overview of the literature is quite nice and could be the basis for a separate manuscript. And despite the large number, referencing is still limited, for example, P5 L4: ". . .fronts has been studied primarily in two-dimensional tanks. . .", recently also 3D patterns observed using geo-electrical imaging (e.g., Ganz et al., VZJ 2014, doi:10.2136/vzj2013.04.0074).

Response: The authors thank the reviewer for the comments. We will reduce the introduction and background section of the manuscript as suggested. We will also take into consideration the reference suggested by the reviewer in order to add information about three-dimensional systems (e.g., Ganz et al., VZJ 2014, doi:10.2136/vzj2013.04.0074).

Furthermore, the specific research hypotheses are not so explicitly stated in the introduction (more indirectly somehow within the review), so that the idea of the experiments and reasons for doing it as it was done remained unclear to me at the end of the introduction, where also the objectives were too general. Clear objectives statements are then found in the discussion and again in the conclusions.

Response: The authors thank the reviewer for the comments. We will revise the section related to the specific research hypotheses and objectives to make them more explicit.

The methods are explained very detailed, for Tables and Figures, however, I found it very difficult to understand without having the abbreviations explained in headers and captions (e.g., Suwanee River Natural Organic Matter (SRNOM) acronym etc).

Response: The authors thank the reviewer for the comments. We will describe the abbreviations in headers and captions of the tables and figures in greater detail, as suggested by the reviewer.

One methodological problem that was probably discussed in earlier papers on the technique (?) was unclear to me. This is how to obtain repeatable uniformly compacted sand samples so that the packing effects are not influencing the effects of the solution composition.

Response: The authors thank the reviewer for the comments. To achieve repeatable uniformly compacted sand in the 2D tank, we used a packing device. We will describe the packing device in more detail in the revised manuscript. A Y-shaped sand loader was used to pour the sand in the tank. The Y-shaped sand loader, containing a piece of cloth, was taped above the tank. A sand volume corresponding to the volume of two tanks was loaded into the sand loader. The cloth was then removed from the tank to create a uniform packing of sand within the tank. The sand remaining in the loader and the loader were then removed. To ensure a consistent density of sand in the tank, a plastic rod was used to tap the top edge of the tank to "settle" the top sand layer.

The stated accuracy of bulk density value (1.5043) with 4 digits is quite ambitious. Wetting and especially the partial wetting during the infiltration may change the arrangement of sand particles such that the pore structure may not be always constant. Although results of all three replicate infiltration experiments are provided, the question whether each of the finger pattern is characteristic for each solution and comparable for the replicate is not clear to me. I like the detailed explanation of results but data analysis seems still a bit limited.

Response: The authors thank the reviewer for the comments. We will correct the accuracy of bulk density value. Following the suggestion of Referee #2 regarding the need for more data analysis and assessment and comparison of the "finger pattern" for the different solutions, as well as the comments of reviewer Dr. Maria Dragila, we will revise the discussion section of our manuscript by including a quantitative analysis of the finger pattern/geometry/dimension, i.e. finger width. We will use the scaling theory of Miller and Miller (1956), applied to finger width (Selker and Schroth, 1998). The results of this quantitative analysis will be presented in a table describing the hydrodynamic scaling of finger width measurements resulting from the infiltration of solutions (i.e., control; citrate 0.1 mg/L, 500mg/L; oxalate 0.1 mg/L, 500 mg/L; tannic acid 0.1 mg/L, 500 mg/L, and organics 0.1 and 10 mg/L) in ASTM graded sand C778. Finger width will be scaled to density, gas-liquid interface tension, density and gas-liquid interface tension of pore water, and square root of the cosine of the contact angle. We will scale the finger width using the square root of the cosine of the contact angle, as Culligan et al. (2005) showed that soil sorptivity is a function of the square root of the cosine of the contact angle. This scaling approach will allow us to analyze the finger geometry and determine the parameters of our systems (i.e., solution and interfacial properties) that most influence the flow phenomena in porous media. As also suggested by Dr. Dragila, we will use the Young-Laplace equation which can be used to express capillary pressure at the pore scale as a function of surface tension and contact angle (Lord et al., 1997). We will make a graph of finger properties with capillary pressure as suggested by Dr. Dragila.

Culligan, P. J., Ivanov, V., Germaine, J. T.: Sorptivity and liquid infiltration into dry soil, Adv. Water Resour. 28, 1010–1020. doi:10.1016/j.advwatres.2005.04.003, 2005.

Miller, E. E., Miller, R. D.: Physical theory for capillary flow phenomena, J. Appl. Phys. doi:10.1063/1.1722370, 1956.

Selker, J. S., Schroth, M. H.: Evaluation of hydrodynamic scaling in porous media using finger dimensions, Water Resour. Res. 34, 1935–1940. doi:10.1029/98wr00625, 1998.

Lord, D. L., Demond, A. H., Salehzadeh, A., Hayes, K. I. M. F.: Influence of Organic Acid Solution Chemistry on Subsurface Transport. 2. Capillary Pressure – Saturation, Environ. Sci. Technol. 31, 2052–2058, 1997.

The hypotheses and how the results could be applied to soils remained unclear.

Response: The authors thank the reviewer for the comments. We will add a discussion about the application of our research to soils.

DETAILED COMMENTS

1. The abstract reads well, I only wondered if the conclusions here correspond to those in the conclusion chapter.

Response: The authors thank the reviewer for the comments. We will check the conclusions of the abstract with the conclusions section of the manuscript to ensure their correspondence and will modify them as necessary.

2. Page 7, Lines 15-25: not necessary and unclear

Response: The authors thank the reviewer for the comments. We will remove these lines.

3. Discussion: Starts with the objectives, first paragraph contains hypothesis and should appear as part of the introduction. I was wondering how was equation 2 used?

Response: The authors thank the reviewer for the comments. We will move the first

paragraph of the discussion to the introduction section to make our objectives more explicit, as previously suggested by the reviewer. Equation (2) is used to present the modeling equation to predict finger width and introduce the topic of fingered flow. As mentioned above in the overview comments, a quantitative analysis of the observed fingered flow results will be presented in a table describing the hydrodynamic scaling of finger width measurements resulting from the infiltration of solutions.

4. Page 17, Lines 22-30: This is more or less an introduction to the closer topic and the results seem to confirm existing knowledge.

Response: The authors thank the reviewer for the comments. We will move the text of lines 22-30 to the introduction section.

5. Page 19, Lines about 5-11: This is doubling introduction

Response: The authors thank the reviewer for the comments. We will revise those lines to avoid doubling the introduction, and some of the information in them will be moved to the introduction.

6. Conclusions chapter gives more a summary of results than conclusions.

Response: The authors thank the reviewer for his comments. We will revise the conclusions section to be a conclusion rather than a summary of the results.
* * *

---

## Author Comment (AC4) · 24 Nov 2017

**GENERAL COMMENTS**

The work is very interesting, relevant, and contributes to our need for a more complete understanding of how the complex soil physics and biochemistry of the soil zone impacts vadose zone processes. The technical work appears to be very well performed and no concerns are raised. The comments below are all associated with the communication of the work and how it fits into present day understanding and existing literature. A few suggestions are made for additional calculations, figures and tables.

Response: The authors thank Dr. Maria Dragila for reviewing our manuscript and for her comments about our research being "very interesting, relevant, and contributes to our need for a more complete understanding of how the complex soil physics and biochemistry of the soil zone impacts vadose zone processes" and that "The technical work appears to be very well performed and no concerns are raised." We will follow the suggestions of the reviewer. Below we have responded to each of the comments of the reviewer and provided detailed information about the proposed revisions.

OVERVIEW COMMENTS

1. The paper could be shortened. The introduction and background is too extensive. Unless there is an intent of making this paper a review paper on preferential flow, the introduction should be limited to a brief note of the importance of preferential flow and then be focused on the specific background that is relevant to this study, which is the effect of fluid properties on finger flow development, stability, and the presence of different solutes in the rhizosphere.

Response: The authors thank the reviewer for the comments. We will reduce the introduction and background section of the manuscript as suggested by the reviewer.

2. The Bond number is calculated but its relevance is never discussed in the Background or Discussion sections.

Response: The authors thank the reviewer for the comments. We will add text and information related to Bond number in background and discussion sections.

3. There is too much emphasis on the statistics of the contact angle and surface tension. The number of figures and the discussion can be reduced. This work is about how these two parameters affect finger flow development, and does not an attempt to characterize the relationship between interfacial properties and concentration for different solutions. The information on contact angle, surface tension, fluid densities, Bond number, etc. can be captured in a table summarizing fluid properties (I realize

that this is in separate tables at present), eliminating the need for figures 6 and 8. The graphs showing time dependent values of contact angle and surface tension are useful.

Response: The authors thank the reviewer for the comments. We will reduce the text related to statistics of the contact angle and surface tension. We respectfully disagree with the reviewer concerning figures 6 and 8, as these figures provide percentage reductions of contact angle and surface tension which we believe are valuable to show; thus we will keep these figures. We will keep the graphs showing contact angle and surface tension versus time as suggested by the reviewer. We will combine Table 3, Table 4, and data from Table 5 in a new table following the suggestion of the reviewer. We will be removing Table 5 and incorporating Bond number calculations and the average density values of Table 5 into a new table that will results from merging Table 3 and Table 4.

4.The discussion on finger geometry (or morphology), saturation overshoot, finger width and moisture content distribution needs some improvement. For example, (A) Finger properties are compared to contact angle values and solution concentrations, but a more complete analysis would be to compare finger geometry to capillary pressure at the wetting front. This may solve the problem that tannic acid (whose contact angle does not change with concentration) contradicts the apparent general result. (B) A graph of finger properties with capillary pressure (using the Young-Laplace equation) which takes both contact angle and surface tension into account would rapidly demonstrate in a visual fashion any trends seen.

Response: The authors thank the reviewer for the comments. Following the reviewer's suggestion, we will revise the discussion section by including a quantitative analysis of the finger geometry/dimension, i.e. finger width. We will use the scaling theory of Miller and Miller (1956), applied to finger width (Selker and Schroth, 1998). The results of this quantitative analysis will be presented in a table describing the hydrodynamic scaling of finger width measurements resulting from the infiltration of solutions (i.e., control; citrate 0.1 mg/L, 500mg/L; oxalate 0.1 mg/L, 500 mg/L; tannic acid 0.1 mg/L,

500 mg/L, and organics 0.1 and 10 mg/L) in ASTM graded sand C778. Finger width will be scaled to density, gas-liquid interface tension, density and gas-liquid interface tension of pore water, and square root of the cosine of the contact angle. We will scale finger width using the square root of the cosine of the contact angle, as Culligan et al. (2005) showed that soil sorptivity is a function of the square root of the cosine of the contact angle This scaling approach will allow us to analyze the finger geometry and determine the parameters of our systems that most influence the flow phenomena in porous media. As suggested by the reviewer, we will also use the Young-Laplace equation which can be used to express capillary pressure at the pore scale as a function of surface tension and contact angle (Lord et al., 1997). We will make a graph of finger properties with capillary pressure as suggested by the reviewer.

Culligan, P. J., Ivanov, V., Germaine, J. T.: Sorptivity and liquid infiltration into dry soil, Adv. Water Resour. 28, 1010–1020. doi:10.1016/j.advwatres.2005.04.003, 2005.

Miller, E. E., Miller, R. D.: Physical theory for capillary flow phenomena, J. Appl. Phys. doi:10.1063/1.1722370, 1956.

Selker, J. S., Schroth, M. H.: Evaluation of hydrodynamic scaling in porous media using finger dimensions, Water Resour. Res. 34, 1935–1940. doi:10.1029/98wr00625, 1998.

Lord, D. L., Demond, A. H., Salehzadeh, A., Hayes, K. I. M. F.: Influence of Organic Acid Solution Chemistry on Subsurface Transport. 2. Capillary Pressure – Saturation, Environ. Sci. Technol. 31, 2052–2058, 1997.

5.The technical work, results and data are valuable but the discussion and conclusions are weak. Some of the concepts associated with capillarity and sorptivity and missstated.

Response: The authors thank the reviewer for the comments. As described in section 4 above, we will perform two additional analyses of our data: hydrodynamic scaling of finger (i.e., finger width will be scaled to density, gas-liquid interface tension, density

and gas-liquid interface tension of pore water, and square root of the cosine of the contact angle, and we will use the Young-Laplace equation to express capillary pressure at the pore scale as a function of surface tension and contact angle. The results of these two additional analyses will be added to the discussion section and the conclusion section. Thus, we will strengthen those sections of our manuscript as recommended by the reviewer. We will also check our manuscript to ensure that the concepts associated with capillary and sorptivity are properly used, as recommended by the reviewer.

6. Equation 2 is noted in the Discussion section but not used in the work. Should this equation be moved to the background section? Can it be used to compare observations with predictions?

Response: The authors thank the reviewer for the comments. Equations 1 and 2 are used to compute contact angle and surface tension using the Kruss Easy Drop instrument. We believe that these equations should be listed in the materials and methods section. Also, as previously suggested by the reviewer in section 4, we will use the Young-Laplace equation which can be used to express capillary pressure at the pore scale as a function of surface tension and contact angle (Lord et al., 1997). Making a graph of finger properties with capillary pressure, as suggested by the reviewer, will allow us to characterize the potential effects of the different solutions.

Lord, D. L., Demond, A. H., Salehzadeh, A., Hayes, K. I. M. F.: Influence of Organic Acid Solution Chemistry on Subsurface Transport. 2. Capillary Pressure – Saturation, Environ. Sci. Technol. 31, 2052–2058, 1997.

7. Significant editorial work is necessary before it is ready for publication or public review. A few editorial comments are noted below but the focus of this review is on the technical content.

Response: The authors thank the reviewer for the comments. We will make the necessary editing of our manuscript to ensure that it is ready for publication.

**SPECIFIC COMMENTS**

Abstract

1. Line 27. Should 'processes' be 'properties'?

Response: The authors thank the reviewer for the comments. We will be correcting the text as suggested.

2. Line 28. Change to " . . .the control with a contact angle of 64.5 deg, and a surface tension of 75.75 N/m. . ." There are three problems with the sentence as is: (1) You have not defined the symbol 'theta' as a contact angle yet. (2) even if you had defined it, "64.5 " is not traditionally how it is written but rather it is written " = 64.5". Same changes required on line 29. (3) I thought surface tension units were N/m, not Nm/m.

Response: The authors thank the reviewer for the comments. We will correct the text as suggested. We will use mN/m for the surface tension units.

3. Line 29: "changes" should be "differences". The last sentence is not a complete sentence. Page 2

Response: The authors thank the reviewer for the comments. We will correct the text as suggested.

4. Line 7: change 'element of' to 'contributes to'

Response: The authors thank the reviewer for the comments. We respectfully disagree with the suggestion and plan to keep the sentence in its original form since the meaning would change if we altered the wording as proposed.

5. Line 9: change 'especially' to 'by controlling" and add 'supporting prior to vegetation.'

Response: The authors thank the reviewer for the comments. We will be correcting the text as suggested.

6. Line 10: sentence starting with 'consequently', needs to be move to the end of

this paragraph. As is, 'consequently' is the incorrect word because the previous statement discusses the existence of water and life at the land atmosphere interface, not groundwater issues. You need to add the logical linkage, that in addition to controlling soil moisture and plant life, soil controls deep drainage of infiltrated water, its rate and quality.

Response: The authors thank the reviewer for the comments. We will modify the text to show a logical linkage, using language similar to the suggested example: "that in addition to controlling soil moisture and plant life, soil controls deep drainage of infiltrated water, its rate and quality."

1. Line 22: 'without' should be 'with'

Response: The authors thank the reviewer for the comments. We will correct the text as suggested.

2. Line 23: change 'instance' to 'interaction'

Response: The authors thank the reviewer for the comments. We will correct the text as suggested.

3. Line 25: 4. The soil hydraulic properties (aka, permeability) and the moisture content are properties that 'set the conditions but the instability is caused by the interplay between the two forces, capillarity and gravity.

Response: The authors thank the reviewer for the comments (3 and 4). We will correct the text as suggested.

5. Line 2: "thus" insinuates that the fact that there are two forces leads to instability. IN reality, both stable and unstable regimes are affected by both gravity and capillary forces. It is when the forces are of nearly equal but opposite in magnitude, that small

irregularities, such as small heterogeneities (which are inevitable, even in a practically homogeneous media) will lead to the condition where gravity dominates, at that point the wetting front breakthrough will occur and a finger will begin to form. Re-reading the glass papers may clarify this point. All instability conditions require a trigger to progress into the next favorable stage.

Response: The authors thank the reviewer for the comments. We will revise the text following the comments of the reviewer.

6. The next two paragraphs (starting with 'The non-linearity', and 'In addition) are too extensive since this paper does not discuss how any of the work will affect or change the present understanding of the mathematical or numerical work. I think the authors could move directly from the points made in the first paragraph of page 4, to the fact that change in contact angle will control capillarity and therefore the propensity for instability. Then mention the relevant work associated with this issue, such as the repellency work, etc.

Response: The authors thank the reviewer for the comments. We will revise the text following the comments of the reviewer and reduce the text of the two paragraphs (starting with 'The non-linearity', and 'In addition').

Page 5.

7. The paragraph starting with 'The vadose zone..." I do not think the vadose zone needs to be defined here, again. The instability you are studying is occurring due to plant and soil solutes, so the soil zone is what is relevant.

Response: The authors thank the reviewer for the comments. We will correct the text as suggested.

8. The paragraph starting with 'The plant root. . ." is interesting by not relevant to this paper. The change in repellency caused by the dehydration of the muscilage is

a change in the properties of the solid, not the solute. This paper focuses on the properties of solutes.

Response: The authors thank the reviewer for the comments. We respectfully disagree with the reviewer comment and plan to keep the paragraph in its original form. Plant root, rhizosphere, and rhizodeposits are elements directly related to our study on flow in porous media (especially since we are studying the influence of plant root exudates on the infiltration process) and therefore they clearly need to be a part of the introduction. The description of previous research on plant root, rhizosphere, and rhizodeposits in the introduction is needed, as in some of their findings it is stated the need of the development of imaging technologies to understand the interactions among biological, chemical, and physical processes in the rhizosphere, and as our study uses imaging technology to visualize and measure flow in porous media under the influence of biogeochemical compounds found in the rhizosphere.

9. The paragraph starting with "Indeed" is not necessary. While noting good examples of serious groundwater contamination, these may not be relevant to development of unstable finger flow issues, nor are these associated with plant exudates. In addition, the two examples given, while associated with preferential flow, they are more closely related to fracture flow than unstable finger flow. More relevant to finger flow instability through soil media are agricultural contaminants.

Response: The authors thank the reviewer for the comments. We will remove this paragraph as suggested by the reviewer.

10. Line 10: regarding the word monitored. The dynamics were not monitored via surface tension or contact angle. The dynamics were monitored via visualization. The fluid properties were quantified by measuring contact angle and surface tension.

Response: The authors thank the reviewer for the comments. We will correct the text as suggested.

11. Line 25: what is the relevance of a hydroponic system?

Response: The authors thank the reviewer for the comments. We will correct the text as suggested and remove "hydroponic systems."

12. Line 25: why was NaCl+HNS rather than DI water used as a control? Not suggesting its wrong, but should explain the rationale.

Response: The authors thank the reviewer for the comments. We will add information in the rationale to clarify why NaCl+HNS rather DI water was used as a control. Basically, NaCl+HNS was used as a control as these compounds are used to simulate the nutrients a plant would need to survive, and the simulated compounds would be present in the rhizosphere.

13. Materials and methods: you state that statistics were performed on the contact angle and surface tension measurements, but there was no mention of how many reps where done.

Response: The authors thank the reviewer for the comments. We will correct the text as suggested by adding the related information.

14. Line 9-14: It is unclear to me how the calibration was performed. How did you measure the moisture content for the different intensities? Or did you preset the moisture content and then measure the intensities? Also, I did not understand the comment about the 80%.

Response: The authors thank the reviewer for the comments. We will correct the text by adding more information and/or clarifying the description related to calibration of the light transmission method for determining degree of saturation in porous media. Also,
we will clarify the comment about 80%.

15. Line 18: what does it mean that the 0% and 100% intensity values "were found"? Did you look for locations where you had 100% and 0% and measure the moisture content there?

Response: The authors thank the reviewer for the comments. We will clarify the text as suggested. Yes, we did look at locations where we had 100% and 0% saturation.

16. Line 22: describe the packing device? Was the sand poured in? How did you avoid horizontal layering of sand during packing?

Response: The authors thank the reviewer for the comments. We will describe the packing device. A Y-shaped sand loader containing a piece of cloth was taped above the tank and was used to pour the sand into the tank. A sand volume corresponding to the volume of two tanks was loaded into the sand loader. The cloth was then removed to create a uniform packing of sand within the tank. The sand remaining in the loader and the loader were then removed. To ensure a consistent density of sand in the tank, a plastic rod was used to tap the top edge of the tank to "settle" the top sand layer.

17. Line 14: the "flow velocity" is the velocity of what? The fluid in the peristaltic pump line, or the raindrop speed, or the surface area averaged input flux rate?

Response: The authors thank the reviewer for the comments. We will clarify the text about the flow velocity.

18. Line 29: dynamic contact angle: the term "dynamic" contact angle is usually associated with the dependence of the contact angle on interface velocity, or in a larger scale the wetting front velocity, but fig 5 it is graphed per unit time, so is this the time-dependent contact angle or the dynamic contact angle?

Response: The authors thank the reviewer for the comments. We will correct the text

as suggested. It is the time-dependent contact angle.

19. Line 9: delete the extra "for"

Response: The authors thank the reviewer for the comments. We will correct the text as suggested.

20. You report contact angle values through time up to 90 seconds, and discuss in detail how much these angles have changes over time. But, you do not discuss the significance of these changes to the dynamics of the finger flow system.

Response: The authors thank the reviewer for the comments. We will discuss the potential significance of these changes to the dynamics of the finger flow system and add text accordingly in the discussion section.

21. Line 19: is this "dynamic" or "time dependent"?

Response: The authors thank the reviewer for the comment. We will correct the text as suggested. It is the time-dependent surface tension.

22. Line 22: add "over time" following the word 'tension'

Response: The authors thank the reviewer for the comments. We will correct the text as suggested.

23. Line 23 and 24: the word "changes" should be "differences"

Response: The authors thank the reviewer for the comment. We will correct the text as suggested.

24. Line 26: delete the word 'respectively'

Response: The authors thank the reviewer for the comments. We will correct the text as suggested.

25. Line 28: sentence needs clarifying that the 'groups' are of surface tension values.

Response: The authors thank the reviewer for the comments. We will clarify the existing text: "The statistical analysis revealed two distinct groups; one consisting of the low concentration of oxalate and tannic acid and the high concentration of citrate and SRNOM; and the other group consisting of the remainder, seen in Figure 8" to clearly describe the other group.

26. Line 9-11: since none of your graphs or images use the terms "plant constituents" or "soil constituents" you should either include their proper names in parenthesis to help the reader.

Response: The authors thank the reviewer for the comments. We will clarify the text as suggested.

27. Lines 13-26: can this be put into a table format?

Response: The authors thank the reviewer for the comments. This text describes the key points of Table 3. We will add a reference to Table 3 at the beginning of the section for clarification. As mentioned previously, we plan to merge the Table 3 with Table 4 and Table 5.

28. Liens 1-19: It may help to illustrate the differences if the data was grouped according to contact angle, surface tension or Bond Number groups rather than chemistry, since it is those properties that should control finger formation.

Response: The authors thank the reviewer for the comments. We respectfully disagree with the reviewer comments about the "grouping" and plan to keep this section in its original form to present the results according to the types of biogeochemical compounds selected in our study which focuses especially on the various biogeochemical compounds.

29. Line 21-31: can this information be displayed in graphical form? The changes in finger saturation cross-section are interesting. Again, it would be interesting to see if there is a grouping according to chemistry concentration, or interfacial properties.

Response: The authors thank the reviewer for the comments. We respectfully disagree with the reviewer comments about displaying this information in a graphical form and plan to keep this section in its original form. The horizontal finger saturation profiles are available in Figures 11 and 12, and the percentages in the text are derived from those profiles. We respectfully disagree with the reviewer comments about the "grouping" and plan to keep this section in its original form to present the results according to the types of biogeochemical compounds selected in our study which focuses especially on the various biogeochemical compounds.

Page 16:

31. Line 27: you mention the intrinsic sorptivity but there is no linkage to the relevancy of this statement to the work presented here. It is true that the intrinsic sorptivity is independent of fluid properties because it is a normalized property.

Response: The authors thank the reviewer for the comments. We will remove this text.

32. What is the relevancy of Equation 2. It was not used in the work? It would be valuable to see how the predictions based on this equation compare to the observations.

Response: The authors thank the reviewer for the comments. This equation is used to present the modeling equation to predict finger width and to introduce the topic of discussion about fingered flow. As mentioned earlier in section 4 of the overview comments of this document, a quantitative analysis of the observed fingered flow results will be presented in a table that will describe the hydrodynamic scaling of finger width measurements resulting from the infiltration of solutions.

33. Top paragraph needs to be improved for clarity. The statement "reduction in sorptivity" and "decrease in contact angle" suggests that it was reduced from some earlier value. But in truth the reason there is overshoot is that the sorptivity at the front is lower than the hydraulic conductivity of the media above it. At the wetting front the advancing contact angle is greater, and therefore the sum of capillary and gravitational forces results in lower rate of forward motion. The pressure overshoot is simply associated with hydraulic pressure of water building up above the (way too slow) wetting front. It is also very likely that what is being seen at the front is the 'dynamic contact angle" which is a contact angle that is even greater than the static contact angle measured at 90 sec mark. The dynamic effect is caused by a competition between inertia, motion below the interface, the flexibility of the interface and the movement (stickiness) of the triple line. The contact angle changes in value with the speed of the wetting front. While it may be valuable to measure the dynamic contact angle of these substances, it is not an easy measurement to take. However, you may be able to obtain it from data you already have since there are a few theoretical relationships that can approximate the value of the dynamic contact angle based on the Capillary Number, the static contact angle and the velocity. There is a relationship between the Capillary Number and the Bond Number.

Response: The authors thank the reviewer for the comments. We will improve the paragraph for clarity as suggested by the reviewer. As previously suggested by the reviewer, we will also use the Young-Laplace equation which can be used to express capillary pressure at the pore scale as a function of surface tension and contact angle (Lord et al., 1997). We will make a graph of finger properties with capillary pressure as suggested by the reviewer.

Lord, D. L., Demond, A. H., Salehzadeh, A., Hayes, K. I. M. F.: Influence of Organic Acid Solution Chemistry on Subsurface Transport. 2. Capillary Pressure – Saturation, Environ. Sci. Technol. 31, 2052–2058, 1997.

34. Line 11: why would the contact angle be close to zero during the initial stages?

Response: The authors thank the reviewer for the comments. In the initial stages of the infiltration process, the wetting front is wetting the sand porous media, making it quite wet or moist; therefore, the contact angle is close to zero under these conditions. We will revise the text to make it clearer.

35. Lines14-16: This sentence is confusing. The way it's written, the first half appears to contradict the second half.

Response: The authors thank the reviewer for the comments. We will clarify the sentence.

36. Line 1: use "differs" instead of "changes"

Response: The authors thank the reviewer for the comments. We will correct the text as suggested.

37. Line 2: change "simulated . . ..concentrations" to "solution chemistry and concentration."

Response: The authors thank the reviewer for the comments. We will correct the text as suggested.

38. Line 4-5: The sentence is unclear. Also, you note a relationship between finger morphology and matric potential of the media. Did you measure the matric potential?

Response: The authors thank the reviewer for the comments. We will clarify the text as suggested.

39. First paragraph: There is quite a bit of repetition with two previous paragraphs where the overshoot mechanisms are also described.

Response: The authors thank the reviewer for the comments. We respectfully disagree with the reviewer comment about the repetition and plan to keep this section in its

original form. The two previous paragraphs discussed the infiltration processes and this section discusses the redistribution processes.

40. Line 13: What is meant by "wettability changes in the flow patterns", do you mean how the wettability changes the flow pattern? Or that the wettability is changing within the flow pattern?

Response: The authors thank the reviewer for the comments. We will revise the sentence to clarify the meaning of "wettability changes in the flow patterns."

41. Line 16: what is meant by "present themselves as" do you mean "result in" ?

Response: The authors thank the reviewer for the comments. We will correct the text as suggested.

42. Line 21, add "(except for Tannic acid)" following the word 'angles'

Response: The authors thank the reviewer for the comments. We will correct the text as suggested.

43. Line 22: by 'prevalent' do you mean 'greater'?

Response: The authors thank the reviewer for the comments. We will correct the text as suggested.

44. Line 22-24: I think that the comparison needs to be made between finger geometry and capillary potential (using Young-Laplace eq), not individually with contact angle. The reason for this is that you will note that tannic acid did not show a difference in contact angle between low and high concentrations, but it did show a large difference in surface tension, and in finger geometry. The capillary pressure calculation would capture that difference. This applies through line 27 in this paragraph.

Response: The authors thank the reviewer for the comments. As discussed in section 4 of the overview comments, we will also use the Young-Laplace equation which can be used to express capillary pressure at the pore scale as a function of surface tension

and contact angle. We will then compare the measured finger geometry (width) with the calculated capillary pressure using the Young-Laplace equation. Results of this analyses will be presented in a graph.

45. Line 1: Describe the behavior observed by Bashir etc., so that the reader does not have to go to that citation to understand what is being discussed.

Response: The authors thank the reviewer for the comments. We will correct the text as suggested by describing in more detail the behavior observed in the studies of Bashir et al. (2011) and Henry and Smith (2002).

46. Line 5: the media's hydraulic properties do not change, it is the fluid's properties that are changing.

Response: The authors thank the reviewer for the comments. We will correct the text following the comments of the reviewer.

47. Line 12: I do not follow how the research presented in this paper is relevant to a discussion associated with the apparent repellency effect of dry mucilage. The study is of a wetting experiment onto dry mineral base (not organics), with a well saturated organic in solution.

Response: The authors thank the reviewer for the comments. This manuscript presents research related to biogeochemical compounds, including the rhizodeposits (e.g., plant exudates). Mucilage is also a rhizodeposit, and therefore we discuss some of our results using a comparison to the known effects of mucilage.

48. Line 13-21: Section needs restructuring. What is being said is unclear. Wrong words are being modified, and some of the thoughts are incomplete.

Response: The authors thank the reviewer for the comments. We will correct the text following the comments of the reviewer by restructuring this section and improving its

clarity.

49. Paragraph starting on line 23 needs sharpening. (A) Line 23-25: sentence needs restructuring. (B) Line 26: mucilage is a type of root exudate. (C)

Response: The authors thank the reviewer for the comments. We will correct the text following the comments of the reviewer by restructuring this section and improving its clarity.

Line 27: what is meant by "stabilizing"? mucilage does act as a glue, and it does create interstitial spaces but it also absorbs water by osmotic processes.

Response: The authors thank the reviewer for the comments. We will correct the text following the comments of the reviewer by clarifying the term "stabilizing" effect that we used to describe the research findings of Walker et al. (2003).

50. Line 16: change 'special' to 'spatial'

Response: The authors thank the reviewer for the comments. We will correct the text as suggested.

51. Line 26-27: change "both. . .constituent" to "solution chemistry and concentration". Change "when compared to the control" to "and the resulting infiltration profile"

Response: The authors thank the reviewer for the comments. We will correct the text as suggested.

52. Line 28: No changes in what?

Response: The authors thank the reviewer for the comments. We will correct the text by clarifying "no changes."

53. Line 2: What is meant by "more normalized"

Response: The authors thank the reviewer for the comments. We will correct the text by removing the term "more normalized."

54. Line 7: delete "or anything", this paper is focused on fluids.

Response: The authors thank the reviewer for the comments. We will correct the text as suggested.

55. Line 8: do you mean efficiency or effectiveness?

Response: The authors thank the reviewer for the comments. We will correct the text as suggested.

---

## Author Response (AR1)

To the authors: Please note that the "SC1: Technical Review" by Maria I Dragila should be treated as a formal solicited review, and so you are also requested to reply to this comment with a detailed response letter, along with the other two formal reviews ("RC1, RC2"). Thank you.

**Response:** The authors thank Editor Brian Berkowitz for his comments. We have treated the "SC1: Technical Review" by Maria I. Dragila as a formal solicited review and provided a detailed response letter, along with our replies to the other two formal reviews ("RC1, RC2").
This paper focuses on the influence of the soil rhizosphere's plant exudates on soil wettability and infiltration. Specifically, the authors study the spatial and temporal development of unstable flow leading to preferential flow "fingering" and their relationship with those plant exudates and soil solution components. The authors propose that these compounds are responsible for keeping the soil around the roots moist by swelling and adsorbing water, and also to keep the contact between the roots and soil particles. To study the effect on preferential flow, the authors employ a 2D tank filled with silica sand equipped with a rainfall simulator on the top of the apparatus in order to create the necessary flow conditions for the infiltration experiments. On the back of the experimental device, a light panel was place so the development of such "fingers" could be track by the so-called light transmission method (LTM). They also measured the contact angle and surface tension of the solutions in the soil by using a Kruss easy drop SA1. To simulate plant exudates they selected sodium citrate and sodium oxalate, as soil com- ponents they used Suwanee River Natural Organic matter (SRNOM) and tannic acid, and as a base solution NaCl+ a nutrient solution. Through these series of experiments, they analyzed (i) the number of fingers formed, (ii) the velocity of their propagation, and (iii) the vertical and horizontal water saturation profiles. This rich data set is used to test main relationships among water chemistry, the porous media properties and the flow dynamics of each individual solution and its effect on the soil infiltration process.

The manuscript is well posed and written and of interest to HESS readers. It is novel as it addresses an important topic for which there is limited knowledge to date. The work is consequential; if plant roots promote the creation of unstable soil water flow, many soil remediation applications will need to be reassessed. Previous literature has theorized these effects but none (or at least none to this reviewer knowledge) studied them experimentally. For instance, the work shows how the addition of surfactant exudates into the soil can alter the solution chemistry enough to produce a change in behavior of the porous media's hydraulic properties, by exudates increase wettability and thus mobility of the soil solution in the porous media. I suggest minor revisions listed below.

**Response:** The authors thank Referee #1 for reviewing our manuscript and for the comments about our manuscript being "well posed and written and of interest to HESS readers" and "novel as it addresses an important topic for which there is limited knowledge to date." The authors also thank the referee for the comments describing our research as: "The work is consequential; if plant roots promote the creation of unstable soil water flow, many soil remediation applications will need to be reassessed. Previous literature has theorized these effects but none (or at least none to this reviewer knowledge) studied them experimentally."

SPECIFIC COMMENTS

Ln 12: Possible typo: 'though' instead of 'through' Ln 16: Possible typo: 'special' instead of 'spatial'

**Response:** The authors thank the reviewer for the comments. We corrected the text as suggested for Ln 12 and Ln 16.

Figures 5 and 6: Although the results are well presented in their corresponding tables, in the figures 5 and 6 it is very difficult to differentiate between the different components and their respective deviations.

**Response:** The authors thank the reviewer for the comments. We used colors to help to distinguish among the different components and their respective deviations in these figures 5 and 6. We also used colors for figures 7 and 8.
The manuscript presents experimental results of unstable flow patterns in sand boxes using a light transmission technique that allows identifying differences in flow patterns caused by the composition of the solution of the irrigation water. Plant exudates and soil solutions with different contact angles and surface tensions were tested and related to the effects on the flow finger development. Results demonstrate quite different patterns during the infiltration process. The test comparing various solution compositions is new and the experiments are highly sophisticated and carefully carried out, especially the combination with the light transmission method that allows determining local water contents is innovative. But the manuscript could be better structured, shortened, and more focused on the analysis of these experiments.

**Response:** The authors thank Referee #2 for reviewing our manuscript and for the comments about our research: "Results demonstrate quite different patterns during the infiltration process. The test comparing various solution compositions is new and the experiments are highly sophisticated and carefully carried out, especially the combination with the light transmission method that allows determining local water contents is innovative." We revised the text of the manuscript and followed the reviewer's suggestions for our manuscript to "be better structured, shortened, and more focused on the analysis of these experiments." We responded below to each of the reviewer's comments and provided detailed information about the revisions.

GENERAL COMMENTS

Claiming this manuscript to be on original research, my immediate impression was that authors should come to the main points more quickly; many references are not further used for the idea and results of this study. When continuing reading, the review part appears more and more excessive; in particular, the multiple referencing is changing the appearance towards a review article in which authors are trying to collect all relevant papers. Such an overview of the literature is quite nice and could be the basis for a separate manuscript. And despite the large number, referencing is still limited, for example, P5 L4: ". . .fronts has been studied primarily in two-dimensional tanks. . .", recently also 3D patterns observed using geo-electrical imaging (e.g., Ganz et al., VZJ 2014, doi:10.2136/vzj2013.04.0074).

**Response:** The authors thank the reviewer for the comments. We reduced the introduction and background section of the manuscript as suggested. We added the reference suggested by the reviewer in order to add information about three-dimensional systems (e.g., Ganz et al., VZJ 2014, doi:10.2136/vzj2013.04.0074).

Furthermore, the specific research hypotheses are not so explicitly stated in the introduction (more indirectly somehow within the review), so that the idea of the experiments and reasons for doing it as it was done remained unclear to me at the end of the introduction, where also the objectives were too general. Clear objectives statements are then found in the discussion and again in the conclusions.

**Response:** The authors thank the reviewer for the comments. We revised the section related to the specific research hypotheses and objectives to make them more explicit.

The methods are explained very detailed, for Tables and Figures, however, I found it very difficult to understand without having the abbreviations explained in headers and captions (e.g., Suwanee River Natural Organic Matter (SRNOM) acronym etc).

**Response:** The authors thank the reviewer for the comments. We described the abbreviations in headers and captions of the tables and figures in greater detail, as suggested by the reviewer.

One methodological problem that was probably discussed in earlier papers on the technique (?) was unclear to me. This is how to obtain repeatable uniformly compacted sand samples so that the packing effects are not influencing the effects of the solution composition.

**Response:** The authors thank the reviewer for the comments. To achieve repeatable uniformly compacted sand in the 2D tank, we used a packing device. We described the packing device in more detail in the revised manuscript. A Y-shaped sand loader was used to pour the sand in the tank. The Y-shaped sand loader, containing a piece of cloth, was taped above the tank. A sand volume corresponding to the volume of two tanks was loaded into the sand loader. The cloth was then removed from the tank to create a uniform packing of sand within the tank. The sand remaining in the loader and the loader were then removed. To ensure a consistent density of sand in the tank, a plastic rod was used to tap the top edge of the tank to "settle" the top sand layer.

The stated accuracy of bulk density value (1.5043) with 4 digits is quite ambitious. Wetting and especially the partial wetting during the infiltration may change the arrangement of sand particles such that the pore structure may not be always constant. Although results of all three replicate infiltration experiments are provided, the question whether each of the finger pattern is characteristic for each solution and comparable for the replicate is not clear to me. I like the detailed explanation of results but data analysis seems still a bit limited.

**Response:** The authors thank the reviewer for the comments. We corrected the accuracy of bulk density value. Following the suggestion of Referee #2 regarding the need for more data analysis and assessment and comparison of the "finger pattern" for the different solutions, as well as the comments of reviewer Dr. Maria Dragila, we revised the discussion section of our manuscript by including a quantitative analysis of the finger pattern/geometry/dimension, i.e. finger width. We used the scaling theory of Miller and Miller (1956), applied to finger width (Selker and Schroth, 1998). The results of this quantitative analysis are presented in a table describing the hydrodynamic scaling of finger width measurements resulting from the infiltration of solutions (i.e., control; citrate 0.1 mg/L, 500mg/L; oxalate 0.1 mg/L, 500 mg/L; tannic acid 0.1 mg/L, 500 mg/L, and organics 0.1 and 10 mg/L) in ASTM graded sand C778. Finger width were scaled to density, gas-liquid interface tension, density and gas-liquid interface tension of pore water, and square root of the cosine of the contact angle. We also scaled the finger width using the square root of the cosine of the contact angle, as Culligan et al. (2005) showed that soil sorptivity is a function of the square root of the cosine of the contact angle. This scaling approach allowed us to analyze the finger geometry and determine the parameters of our systems (i.e., solution and interfacial properties) that most influence the flow phenomena in porous media. As also suggested by Dr. Dragila, we used the Young-Laplace equation which can be used to express capillary pressure at the pore scale as a function of surface tension and contact angle (Lord et al., 1997). We made a graph of finger properties with capillary pressure as suggested by Dr. Dragila.

Culligan, P. J., Ivanov, V., Germaine, J. T.: Sorptivity and liquid infiltration into dry soil, Adv. Water Resour. 28, 1010–1020. doi:10.1016/j.advwatres.2005.04.003, 2005.

Miller, E. E., Miller, R. D.: Physical theory for capillary flow phenomena, J. Appl. Phys. doi:10.1063/1.1722370, 1956.

Selker, J. S., Schroth, M. H.: Evaluation of hydrodynamic scaling in porous media using finger dimensions, Water Resour. Res. 34, 1935–1940. doi:10.1029/98wr00625, 1998.

Lord, D. L., Demond, A. H., Salehzadeh, A., Hayes, K. I. M. F.: Influence of Organic Acid Solution Chemistry on Subsurface Transport. 2. Capillary Pressure – Saturation, Environ. Sci. Technol. 31, 2052–2058, 1997.

The hypotheses and how the results could be applied to soils remained unclear.

**Response:** The authors thank the reviewer for the comments. Regarding the comments of the reviewer on "The hypotheses and how the results could be applied to soils remained unclear," this is currently beyond of the scope of this research. Although in the future we plan to investigate fingered flow in natural soils.

DETAILED COMMENTS

1. The abstract reads well, I only wondered if the conclusions here correspond to those in the conclusion chapter.

**Response:** The authors thank the reviewer for the comments. We checked and revised the conclusions of the abstract with the conclusions section of the manuscript to ensure their correspondence and modified them as necessary.

2. Page 7, Lines 15-25: not necessary and unclear

**Response:** The authors thank the reviewer for the comments. We removed these lines.

3. Discussion: Starts with the objectives, first paragraph contains hypothesis and should appear as part of the introduction. I was wondering how was equation 2 used?

**Response:** The authors thank the reviewer for the comments. We moved the first paragraph of the discussion to the introduction section to make our objectives more explicit, as previously suggested by the reviewer. Equation (2) is used to present the modeling equation to predict finger width and introduce the topic of fingered flow. As mentioned above in the overview comments, a quantitative analysis of the observed fingered flow results is presented in a table describing the hydrodynamic scaling of finger width measurements resulting from the infiltration of solutions.

4. Page 17, Lines 22-30: This is more or less an introduction to the closer topic and the results seem to confirm existing knowledge.

**Response:** The authors thank the reviewer for the comments. We moved the text of lines 22-30 to the introduction section.

5. Page 19, Lines about 5-11: This is doubling introduction

**Response:** The authors thank the reviewer for the comments. We revised those lines to avoid doubling the introduction, and some of the information in them will be moved to the introduction.

6. Conclusions chapter gives more a summary of results than conclusions.

**Response:** The authors thank the reviewer for his comments. We revised the conclusions section to be a conclusion rather than a summary of the results.

The work is very interesting, relevant, and contributes to our need for a more complete understanding of how the complex soil physics and biochemistry of the soil zone impacts vadose zone processes. The technical work appears to be very well performed and no concerns are raised. The comments below are all associated with the communication of the work and how it fits into present day understanding and existing literature. A few suggestions are made for additional calculations, figures and tables.

**Response:** The authors thank Dr. Maria Dragila for reviewing our manuscript and for her comments about our research being "very interesting, relevant, and contributes to our need for a more complete understanding of how the complex soil physics and biochemistry of the soil zone impacts vadose zone processes" and that "The technical work appears to be very well performed and no concerns are raised." We followed the suggestions of the reviewer. Below we have responded to each of the comments of the reviewer and provided detailed information about the revisions.

OVERVIEW COMMENTS

1. The paper could be shortened. The introduction and background is too extensive. Unless there is an intent of making this paper a review paper on preferential flow, the introduction should be limited to a brief note of the importance of preferential flow and then be focused on the specific background that is relevant to this study, which is the effect of fluid properties on finger flow development, stability, and the presence of different solutes in the rhizosphere.

**Response:** The authors thank the reviewer for the comments. We have reduced the introduction and background section of the manuscript as suggested by the reviewer.

2. The Bond number is calculated but its relevance is never discussed in the Background or Discussion sections.

**Response:** The authors thank the reviewer for the comments. We removed calculation related to Bond number since we did not use it in our discussion.

3. There is too much emphasis on the statistics of the contact angle and surface tension. The number of figures and the discussion can be reduced. This work is about how these two parameters affect finger flow development, and does not an attempt to characterize the relationship between interfacial properties and concentration for different solutions. The information on contact angle, surface tension, fluid densities, Bond number, etc. can be captured in a table summarizing fluid properties (I realize that this is in separate tables at present), eliminating the need for figures 6 and 8. The graphs showing time dependent values of contact angle and surface tension are useful.

**Response:** The authors thank the reviewer for the comments. We respectfully disagree with the reviewer concerning figures 6 and 8, as these figures provide percentage reductions of contact angle and surface

tension which we believe are valuable to show; thus we kept these figures. We kept the graphs showing contact angle and surface tension versus time as suggested by the reviewer. We combined Table 3, Table 4, and data from Table 5 in a new table following the suggestion of the reviewer. We removed Table 5 and Bond number calculations, and incorporated the average density values of Table 5 into a new table that resulted from merging Table 3 and Table 4.

4.The discussion on finger geometry (or morphology), saturation overshoot, finger width and moisture content distribution needs some improvement. For example, (A) Finger properties are compared to contact angle values and solution concentrations, but a more complete analysis would be to compare finger geometry to capillary pressure at the wetting front. This may solve the problem that tannic acid (whose contact angle does not change with concentration) contradicts the apparent general result. (B) A graph of finger properties with capillary pressure (using the Young-Laplace equation) which takes both contact angle and surface tension into account would rapidly demonstrate in a visual fashion any trends seen.

**Response:** The authors thank the reviewer for the comments. Following the reviewer's suggestion, we revised the discussion section by including a quantitative analysis of the finger geometry/dimension, i.e. finger width. We used the scaling theory of Miller and Miller (1956), applied to finger width (Selker and Schroth, 1998). The results of this quantitative analysis are presented in a table describing the hydrodynamic scaling of finger width measurements resulting from the infiltration of solutions (i.e., control; citrate 0.1 mg/L, 500mg/L; oxalate 0.1 mg/L, 500 mg/L; tannic acid 0.1 mg/L, 500 mg/L, and organics 0.1 and 10 mg/L) in ASTM graded sand C778. Finger width was scaled to density, gas-liquid interface tension, density and gas-liquid interface tension of pore water, and square root of the cosine of the contact angle. We also scaled finger width using the square root of the cosine of the contact angle, as Culligan et al. (2005) showed that soil sorptivity is a function of the square root of the cosine of the contact angle. This scaling approach allowed us to analyze the finger geometry and determine the parameters of our systems that most influence the flow phenomena in porous media. As suggested by the reviewer, we also used the Young-Laplace equation which can be used to express capillary pressure at the pore scale as a function of surface tension and contact angle (Lord et al., 1997). We made a graph of finger properties with capillary pressure as suggested by the reviewer.

Culligan, P. J., Ivanov, V., Germaine, J. T.: Sorptivity and liquid infiltration into dry soil, Adv. Water Resour. 28, 1010–1020. doi:10.1016/j.advwatres.2005.04.003, 2005.

Miller, E. E., Miller, R. D.: Physical theory for capillary flow phenomena, J. Appl. Phys. doi:10.1063/1.1722370, 1956.

Selker, J. S., Schroth, M. H.: Evaluation of hydrodynamic scaling in porous media using finger dimensions, Water Resour. Res. 34, 1935–1940. doi:10.1029/98wr00625, 1998.

Lord, D. L., Demond, A. H., Salehzadeh, A., Hayes, K. I. M. F.: Influence of Organic Acid Solution Chemistry on Subsurface Transport. 2. Capillary Pressure – Saturation, Environ. Sci. Technol. 31, 2052–2058, 1997.

5.The technical work, results and data are valuable but the discussion and conclusions are weak. Some of the concepts associated with capillarity and sorptivity and missstated.

**Response:** The authors thank the reviewer for the comments. As described in section 4 above, we performed two additional analyses of our data: hydrodynamic scaling of finger (i.e., finger width) was scaled to density, gas-liquid interface tension, density and gas-liquid interface tension of pore water, and square root of the cosine of the contact angle, and we used the Young-Laplace equation to express capillary pressure at the pore scale as a function of surface tension and contact angle. The results of these two additional analyses

were added to the discussion section and the conclusion section. Thus, we strengthened those sections of our manuscript as recommended by the reviewer. We also checked our manuscript to ensure that the concepts associated with capillary and sorptivity were properly used, as recommended by the reviewer.

6. Equation 2 is noted in the Discussion section but not used in the work. Should this equation be moved to the background section? Can it be used to compare observations with predictions?

**Response:** The authors thank the reviewer for the comments. Equations 1 and 2 are used to compute contact angle and surface tension using the Kruss Easy Drop instrument. We believe that these equations should be listed in the materials and methods section. Also, as previously suggested by the reviewer in section 4, we used the Young-Laplace equation which can be used to express capillary pressure at the pore scale as a function of surface tension and contact angle (Lord et al., 1997). Making a graph of finger properties with capillary pressure, as suggested by the reviewer, allowed us to characterize the potential effects of the different solutions.

Lord, D. L., Demond, A. H., Salehzadeh, A., Hayes, K. I. M. F.: Influence of Organic Acid Solution Chemistry on Subsurface Transport. 2. Capillary Pressure – Saturation, Environ. Sci. Technol. 31, 2052–2058, 1997.

7. Significant editorial work is necessary before it is ready for publication or public review. A few editorial comments are noted below but the focus of this review is on the technical content.

**Response:** The authors thank the reviewer for the comments. We made the necessary editing of our manuscript to ensure that it is ready for publication.

SPECIFIC COMMENTS

Abstract

1. Line 27. Should 'processes' be 'properties'?

**Response:** The authors thank the reviewer for the comments. We corrected the text as suggested.

2. Line 28. Change to " . . .the control with a contact angle of 64.5 deg, and a surface tension of 75.75 N/m. . ." There are three problems with the sentence as is: (1) You have not defined the symbol 'theta' as a contact angle yet. (2) even if you had defined it, "64.5 " is not traditionally how it is written but rather it is written " = 64.5". Same changes required on line 29. (3) I thought surface tension units were N/m, not Nm/m.

**Response:** The authors thank the reviewer for the comments. We corrected the text as suggested. We used mN/m for the surface tension units.

3. Line 29: "changes" should be "differences". The last sentence is not a complete sentence. Page 2

**Response:** The authors thank the reviewer for the comments. We corrected the text as suggested and revised the abstract.

4. Line 7: change 'element of' to 'contributes to'

**Response:** The authors thank the reviewer for the comments. We respectfully disagree with the suggestion and kept the sentence in its original form since the meaning would change if we altered the wording as proposed.

5. Line 9: change 'especially' to 'by controlling" and add 'supporting prior to vegetation.

**Response:** The authors thank the reviewer for the comments. We corrected the text as suggested.

6. Line 10: sentence starting with 'consequently', needs to be move to the end of this paragraph. As is, 'consequently' is the incorrect word because the previous statement discusses the existence of water and life at the land atmosphere interface, not groundwater issues. You need to add the logical linkage, that in addition to controlling soil moisture and plant life, soil controls deep drainage of infiltrated water, its rate and quality.

**Response:** The authors thank the reviewer for the comments. We modified the text to show a logical linkage, using language similar to the suggested example: "that in addition to controlling soil moisture and plant life, soil controls deep drainage of infiltrated water, its rate and quality."

1. Line 22: 'without' should be 'with'

**Response:** The authors thank the reviewer for the comments. We corrected the text as suggested.

2. Line 23: change 'instance' to 'interaction'

**Response:** The authors thank the reviewer for the comments. We corrected the text as suggested.

3. Line 25:
4. The soil hydraulic properties (aka, permeability) and the moisture content are properties that 'set the conditions but the instability is caused by the interplay between the two forces, capillarity and gravity.

**Response:** The authors thank the reviewer for the comments (3 and 4). We corrected the text as suggested.

5. Line 2: "thus" insinuates that the fact that there are two forces leads to instability. IN reality, both stable and unstable regimes are affected by both gravity and capillary forces. It is when the forces are of nearly equal but opposite in magnitude, that small irregularities, such as small heterogeneities (which are inevitable, even in a practically homogeneous media) will lead to the condition where gravity dominates, at that point the wetting front breakthrough will occur and a finger will begin to form. Re-reading the glass papers may clarify this point. All instability conditions require a trigger to progress into the next favorable stage.

**Response:** The authors thank the reviewer for the comments. We corrected the text following the comments of the reviewer.

6. The next two paragraphs (starting with 'The non-linearity', and 'In addition) are too extensive since this paper does not discuss how any of the work will affect or change the present understanding of the mathematical or numerical work. I think the authors could move directly from the points made in the first

paragraph of page 4, to the fact that change in contact angle will control capillarity and therefore the propensity for instability. Then mention the relevant work associated with this issue, such as the repellency work, etc.

**Response:** The authors thank the reviewer for the comments. We revised the text following the comments of the reviewer and reduce the text of the two paragraphs (starting with 'The non-linearity', and 'In addition').

Page 5.

7. The paragraph starting with 'The vadose zone...” I do not think the vadose zone
needs to be defined here, again. The instability you are studying is occurring due to plant and soil solutes, so the soil zone is what is relevant.

**Response:** The authors thank the reviewer for the comments. We corrected the text as suggested.

8. The paragraph starting with 'The plant root. . .” is interesting by not relevant to this paper. The change in repellency caused by the dehydration of the muscilage is a change in the properties of the solid, not the solute. This paper focuses on the properties of solutes.

**Response:** The authors thank the reviewer for the comments. We respectfully disagree with the reviewer comment and kept the paragraph in its original form. Plant root, rhizosphere, and rhizodeposits are elements directly related to our study on flow in porous media (especially since we are studying the influence of plant root exudates on the infiltration process) and therefore they clearly need to be a part of the introduction. The description of previous research on plant root, rhizosphere, and rhizodeposits in the introduction is needed, as in some of their findings it is stated the need of the development of imaging technologies to understand the interactions among biological, chemical, and physical processes in the rhizosphere, and as our study uses imaging technology to visualize and measure flow in porous media under the influence of biogeochemical compounds found in the rhizosphere.

9. The paragraph starting with "Indeed" is not necessary. While noting good examples of serious groundwater contamination, these may not be relevant to development of unstable finger flow issues, nor are these associated with plant exudates. In addition, the two examples given, while associated with preferential flow, they are more closely related to fracture flow than unstable finger flow. More relevant to finger flow instability through soil media are agricultural contaminants.

**Response:** The authors thank the reviewer for the comments. We removed this paragraph as suggested by the reviewer.

10. Line 10: regarding the word monitored. The dynamics were not monitored via surface tension or contact angle. The dynamics were monitored via visualization. The fluid properties were quantified by measuring contact angle and surface tension.

**Response:** The authors thank the reviewer for the comments. We corrected the text as suggested.

11. Line 25: what is the relevance of a hydroponic system?

**Response:** The authors thank the reviewer for the comments. We corrected the text as suggested and remove "hydroponic systems."

12. Line 25: why was NaCl+HNS rather than DI water used as a control? Not suggesting its wrong, but should explain the rationale.

**Response:** The authors thank the reviewer for the comments. We added information in the rationale to clarify why NaCl+HNS rather DI water was used as a control. Basically, NaCl+HNS was used as a control as these compounds are used to simulate the nutrients a plant needs to survive, and the simulated compounds are present in the rhizosphere.

13. Materials and methods: you state that statistics were performed on the contact angle and surface tension measurements, but there was no mention of how many reps where done.

**Response:** The authors thank the reviewer for the comments. We corrected the text as suggested by adding the related information.

14. Line 9-14: It is unclear to me how the calibration was performed. How did you measure the moisture content for the different intensities? Or did you preset the moisture content and then measure the intensities? Also, I did not understand the comment about the 80%.

**Response:** The authors thank the reviewer for the comments. We corrected the text by adding more information and clarified the description related to calibration of the light transmission method for determining degree of saturation in porous media. Also, we removed the comment about 80%.

15. Line 18: what does it mean that the 0% and 100% intensity values "were found"? Did you look for locations where you had 100% and 0% and measure the moisture content there?

**Response:** The authors thank the reviewer for the comments. We clarified the text as suggested. Yes, we did look at locations where we had 100% and 0% saturation.

16. Line 22: describe the packing device? Was the sand poured in? How did you avoid horizontal layering of sand during packing?

**Response:** The authors thank the reviewer for the comments. We added a description of the packing device. A Y-shaped sand loader containing a piece of cloth was taped above the tank and was used to pour the sand into the tank. A sand volume corresponding to the volume of two tanks was loaded into the sand loader. The cloth was then removed to create a uniform packing of sand within the tank. The sand remaining in the loader and the loader were then removed. To ensure a consistent density of sand in the tank, a plastic rod was used to tap the top edge of the tank to "settle" the top sand layer.

17. Line 14: the "flow velocity" is the velocity of what? The fluid in the peristaltic pump line, or the raindrop speed, or the surface area averaged input flux rate?

**Response:** The authors thank the reviewer for the comments. We clarified the text about the flow velocity.

18. Line 29: dynamic contact angle: the term "dynamic" contact angle is usually associated with the dependence of the contact angle on interface velocity, or in a larger scale the wetting front velocity, but fig 5 it is graphed per unit time, so is this the time- dependent contact angle or the dynamic contact angle?

**Response:** The authors thank the reviewer for the comments. We corrected the text as suggested. It is the time-dependent contact angle.

19. Line 9: delete the extra "for"

**Response:** The authors thank the reviewer for the comments. We corrected the text as suggested.

20. You report contact angle values through time up to 90 seconds, and discuss in detail how much these angles have changes over time. But, you do not discuss the significance of these changes to the dynamics of the finger flow system.

**Response:** The authors thank the reviewer for the comments. This is an interesting comment and it may be the subject of future research.

21. Line 19: is this "dynamic" or "time dependent"?

**Response:** The authors thank the reviewer for the comment. We corrected the text as suggested. It is the time-dependent surface tension.

22. Line 22: add "over time" following the word 'tension'

**Response:** The authors thank the reviewer for the comments. We corrected the text as suggested.

23. Line 23 and 24: the word "changes" should be "differences"

**Response:** The authors thank the reviewer for the comment. We corrected the text as suggested.

24. Line 26: delete the word 'respectively'

**Response:** The authors thank the reviewer for the comments. We corrected the text as suggested.

25. Line 28: sentence needs clarifying that the 'groups' are of surface tension values. Page 14

**Response:** The authors thank the reviewer for the comments. We clarified the existing text: "The statistical analysis revealed two distinct groups; one consisting of the low concentration of oxalate and tannic acid and the high concentration of citrate and SRNOM; and the other group consisting of the remainder, seen in Figure 8" to clearly describe the other group.

26. Line 9-11: since none of your graphs or images use the terms "plant constituents" or "soil constituents" you should either include their proper names in parenthesis to help the reader.

**Response:** The authors thank the reviewer for the comments. We clarified the text as suggested and modified the text of figures and tables as necessary.

27. Lines 13-26: can this be put into a table format?

**Response:** The authors thank the reviewer for the comments. This text describes the key points of Table 3. We added a reference to Table 3 at the beginning of the section for clarification. As mentioned previously, we merged the Table 3 with Table 4 and Table 5.

28. Liens 1-19: It may help to illustrate the differences if the data was grouped according to contact angle, surface tension or Bond Number groups rather than chemistry, since it is those properties that should control finger formation.

**Response:** The authors thank the reviewer for the comments. We respectfully disagree with the reviewer comments about the "grouping" and kept this section in its original form to present the results according to the types of biogeochemical compounds selected in our study which focuses especially on the various biogeochemical compounds.

29. Line 21-31: can this information be displayed in graphical form? The changes in finger saturation cross-section are interesting. Again, it would be interesting to see if there is a grouping according to chemistry concentration, or interfacial properties.

**Response:** The authors thank the reviewer for the comments. We respectfully disagree with the reviewer comments about displaying this information in a graphical form and kept this section in its original form. The horizontal finger saturation profiles are available in Figures 11 and 12, and the percentages in the text are derived from those profiles. We respectfully disagree with the reviewer comments about the "grouping" and kept this section in its original form to present the results according to the types of biogeochemical compounds selected in our study which focuses especially on the various biogeochemical compounds.

Page 16:

31. Line 27: you mention the intrinsic sorptivity but there is no linkage to the relevancy of this statement to the work presented here. It is true that the intrinsic sorptivity is independent of fluid properties because it is a normalized property.

**Response:** The authors thank the reviewer for the comments. We removed this text.

32. What is the relevancy of Equation 2. It was not used in the work? It would be valuable to see how the predictions based on this equation compare to the observations.

**Response:** The authors thank the reviewer for the comments. This equation is used to present the modeling equation to predict finger width and to introduce the topic of discussion about fingered flow. As mentioned earlier in section 4 of the overview comments of this document, a quantitative analysis of the observed fingered flow results has been added and is presented in a table that describes the hydrodynamic scaling of finger width measurements resulting from the infiltration of solutions.

33. Top paragraph needs to be improved for clarity. The statement "reduction in sorptivity" and "decrease in contact angle" suggests that it was reduced from some earlier value. But in truth the reason there is overshoot is that the sorptivity at the front is lower than the hydraulic conductivity of the media above it. At the wetting front the advancing contact angle is greater, and therefore the sum of capillary and

gravitational forces results in lower rate of forward motion. The pressure overshoot is simply associated with hydraulic pressure of water building up above the (way too slow) wetting front. It is also very likely that what is being seen at the front is the 'dynamic contact angle" which is a contact angle that is even greater than the static contact angle measured at 90 sec mark. The dynamic effect is caused by a competition between inertia, motion below the interface, the flexibility of the interface and the movement (stickiness) of the triple line. The contact angle changes in value with the speed of the wetting front. While it may be valuable to measure the dynamic contact angle of these substances, it is not an easy measurement to take. However, you may be able to obtain it from data you already have since there are a few theoretical relationships that can approximate the value of the dynamic contact angle based on the Capillary Number, the static contact angle and the velocity. There is a relationship between the Capillary Number and the Bond Number.

**Response:** The authors thank the reviewer for the comments. We improved the paragraph for clarity as suggested by the reviewer. As previously suggested by the reviewer, we also used the Young-Laplace equation which can be used to express capillary pressure at the pore scale as a function of surface tension and contact angle (Lord et al., 1997). We made a graph of finger properties with capillary pressure as suggested by the reviewer.

Lord, D. L., Demond, A. H., Salehzadeh, A., Hayes, K. I. M. F.: Influence of Organic Acid Solution Chemistry on Subsurface Transport. 2. Capillary Pressure – Saturation, Environ. Sci. Technol. 31, 2052–2058, 1997.

34. Line 11: why would the contact angle be close to zero during the initial stages?

**Response:** The authors thank the reviewer for the comments. In the initial stages of the infiltration process, the wetting front is wetting the sand porous media, making it quite wet or moist; therefore, the contact angle is close to zero under these conditions. We revised the text to make it clearer.

35. Lines14-16: This sentence is confusing. The way it's written, the first half appears to contradict the second half.

**Response:** The authors thank the reviewer for the comments. We clarified the sentence.

36. Line 1: use "differs" instead of "changes"

**Response:** The authors thank the reviewer for the comments. We corrected the text as suggested.

37. Line 2: change "simulated . . ..concentrations" to "solution chemistry and concentration."

**Response:** The authors thank the reviewer for the comments. We corrected the text as suggested.

38. Line 4-5: The sentence is unclear. Also, you note a relationship between finger morphology and matric potential of the media. Did you measure the matric potential?

**Response:** The authors thank the reviewer for the comments. We clarified the text as suggested.

39. First paragraph: There is quite a bit of repetition with two previous paragraphs where the overshoot mechanisms are also described.

**Response:** The authors thank the reviewer for the comments. We respectfully disagree with the reviewer comment about the repetition and kept this section in its original form. The two previous paragraphs discussed the infiltration processes and this section discusses the redistribution processes.

40. Line 13: What is meant by "wettability changes in the flow patterns", do you mean how the wettability changes the flow pattern? Or that the wettability is changing within the flow pattern?

**Response:** The authors thank the reviewer for the comments. We revised the sentence to clarify the meaning of "wettability changes in the flow patterns."

41. Line 16: what is meant by "present themselves as" do you mean "result in" ?

**Response:** The authors thank the reviewer for the comments. We corrected the text as suggested.

42. Line 21, add "(except for Tannic acid)" following the word 'angles'

**Response:** The authors thank the reviewer for the comments. We corrected the text as suggested.

43. Line 22: by 'prevalent' do you mean 'greater'?

**Response:** The authors thank the reviewer for the comments. We corrected the text as suggested.

44. Line 22-24: I think that the comparison needs to be made between finger geometry and capillary potential (using Young-Laplace eq), not individually with contact angle. The reason for this is that you will note that tannic acid did not show a difference in contact angle between low and high concentrations, but it did show a large difference in surface tension, and in finger geometry. The capillary pressure calculation would capture that difference. This applies through line 27 in this paragraph.

**Response:** The authors thank the reviewer for the comments. As discussed in section 4 of the overview comments, we used the Young-Laplace equation which can be used to express capillary pressure at the pore scale as a function of surface tension and contact angle. We compared the measured finger geometry (width) with the calculated capillary pressure using the Young-Laplace equation. Results of this analyses are presented in a graph.

45. Line 1: Describe the behavior observed by Bashir etc., so that the reader does not have to go to that citation to understand what is being discussed.

**Response:** The authors thank the reviewer for the comments. We corrected the text as suggested by describing in more detail the behavior observed in the studies of Bashir et al. (2011) and Henry and Smith (2002).

46. Line 5: the media's hydraulic properties do not change, it is the fluid's properties that are changing.

**Response:** The authors thank the reviewer for the comments. We corrected the text following the comments of the reviewer.

47. Line 12: I do not follow how the research presented in this paper is relevant to a discussion associated with the apparent repellency effect of dry mucilage. The study is of a wetting experiment onto dry mineral base (not organics), with a well saturated organic in solution.

**Response:** The authors thank the reviewer for the comments. This manuscript presents research related to biogeochemical compounds, including the rhizodeposits (e.g., plant exudates). Mucilage is also a rhizodeposit, and therefore we discuss some of our results using a comparison to the known effects of mucilage.

48. Line 13-21: Section needs restructuring. What is being said is unclear. Wrong words are being modified, and some of the thoughts are incomplete.

**Response:** The authors thank the reviewer for the comments. We corrected the text following the comments of the reviewer by restructuring this section and improving its clarity.

49. Paragraph starting on line 23 needs sharpening. (A) Line 23-25: sentence needs restructuring. (B) Line 26: mucilage is a type of root exudate. (C)

**Response:** The authors thank the reviewer for the comments. We corrected the text following the comments of the reviewer by restructuring this section and improving its clarity.

Line 27: what is
meant by "stabilizing"? mucilage does act as a glue, and it does create interstitial spaces but it also absorbs water by osmotic processes.

**Response:** The authors thank the reviewer for the comments. We corrected the text following the comments of the reviewer by clarifying the term "stabilizing" effect that we used to describe the research findings of Walker et al. (2003).

50. Line 16: change 'special' to 'spatial'

**Response:** The authors thank the reviewer for the comments. We corrected the text as suggested.

51. Line 26-27: change "both. . .constituent" to "solution chemistry and concentration". Change "when compared to the control" to "and the resulting infiltration profile"

**Response:** The authors thank the reviewer for the comments. We corrected the text as suggested.

52. Line 28: No changes in what?

**Response:** The authors thank the reviewer for the comments. We have revised our conclusion.

53. Line 2: What is meant by "more normalized"

**Response:** The authors thank the reviewer for the comments. We have revised our conclusion section.

54. Line 7: delete "or anything", this paper is focused on fluids.

**Response:** The authors thank the reviewer for the comments. We have revised our conclusion section.

55. Line 8: do you mean efficiency or effectiveness?

**Response:** The authors thank the reviewer for the comments. We have revised our conclusion section.

[revised manuscript text omitted]
 | $64.5 \pm 2.6^A$ | $75.8 \pm 0.5^A$ | 600 | $2.5^C$ | $0.029 \pm 0.003^{AB}$ | $6.13 \pm 2.63^{AB}$ | $17.32 \pm 1.67^{AB}$ | 9.30 | 8.52 | 7.41 | 5.67 | $7.73 \pm 1.1$ |
| Citrate 0.1 mg/L | $52.6 \pm 2.3^D$ | $70.8 \pm 0.4^B$ | 600 | $1.5^B$ | $0.037 \pm 0.003^{AB}$ | $3.50 \pm 1.00^{AB}$ | $22.36 \pm 1.72^{AB}$ | 6.83 | 5.71 | 6.42 | 5.59 | $6.14 \pm 0.6$ |
| Citrate 500 mg/L | $46 \pm 4.9^{BC}$ | $73.6 \pm 1.1^C$ | 600 | $2.0^B$ | $0.032 \pm 0.011^{AB}$ | $2.50 \pm 1.50^{AB}$ | $19.46 \pm 6.63^{AB}$ | 9.23 | 7.56 | 6.68 | 5.50 | $7.24 \pm 1.3$ |
| Oxalate 0.1 mg/L | $54.8 \pm 0.6^{CD}$ | $73.1 \pm 0.4^C$ | 600 | $2.5^B$ | $0.034 \pm 0.005^{AB}$ | $3.38 \pm 1.88^{AB}$ | $20.24 \pm 2.94^{AB}$ | 7.50 | 7.90 | 7.89 | 6.31 | $7.40 \pm 0.3$ |
| Oxalate 500 mg/L | $47.3 \pm 2.3^B$ | $71.6 \pm 0.6^B$ | 600 | $2.5^A$ | $0.037 \pm 0.005^{AB}$ | $2.25 \pm 0.25^B$ | $22.16 \pm 2.88^{AB}$ | 8.41 | 6.55 | 6.77 | 6.13 | $6.97 \pm 0.6$ |
| Tannic Acid 0.1 mg/L | $52.3 \pm 1.2^B$ | $72.7 \pm 0.4^C$ | 600 | $2.0^B$ | $0.032 \pm 0.003^{AB}$ | $4.25 \pm 0.00^{AB}$ | $19.14 \pm 1.64^{AB}$ | 10.35 | 9.11 | 6.71 | 5.11 | $7.82 \pm 0.32^A$ |
| Tannic Acid 500 mg/L | $54.4 \pm 2.7^{BC}$ | $70.8 \pm 0.3^B$ | 600 | $2.0^B$ | $0.042 \pm 0.008^A$ | $2.50 \pm 0.50^{AB}$ | $25.49 \pm 4.87^A$ | 5.78 | 6.79 | 8.67 | 8.71 | $7.49 \pm 0.63^A$ |
| Organics (SRNOM) 0.1 mg/L | $61.7 \pm 2.6^D$ | $71.1 \pm 0.2^B$ | 600 | $2.5^B$ | $0.029 \pm 0.001^{AB}$ | $7.00 \pm 2.00^A$ | $17.66 \pm 0.81^B$ | 7.13 | 7.15 | 7.40 | 4.79 | $6.62 \pm 0.38^A$ |
| Organics (SRNOM) 10 mg/L | $45.0 \pm 2.6^A$ | $73.4 \pm 0.4^C$ | 600 | $2.0^B$ | $0.022 \pm 0.000^B$ | $3.25 \pm 0.75^{AB}$ | $13.29 \pm 0.15^A$ | 6.30 | 5.49 | 6.31 | 5.89 | $6.00 \pm 0.16^A$ |
| **Fully Formed Finger Average** | | | | | | | | | | | | |
| NaCl+HNS | $64.5 \pm 2.6^A$ | $75.8 \pm 0.5^A$ | 945 | $2.5^A$ | $0.031 \pm 0.004^B$ | $8.13 \pm 0.63^{AB}$ | $28.84 \pm 1.25^A$ | 10.05 | 9.06 | 9.53 | 8.30 | $9.24 \pm 0.84^{AB}$ |
| Citrate 0.1 mg/L | $52.6 \pm 2.3^D$ | $70.8 \pm 0.4^B$ | 750 | $1.5^A$ | $0.039 \pm 0.002^{AB}$ | $3.50 \pm 1.00^{BC}$ | $29.46 \pm 0.64^A$ | 6.99 | 6.75 | 6.87 | 5.93 | $6.63 \pm 0.67^C$ |
| Citrate 500 mg/L | $46 \pm 4.9^{BC}$ | $73.6 \pm 1.1^C$ | 840 | $2.0^A$ | $0.037 \pm 0.008^{AB}$ | $3.88 \pm 2.88^{BC}$ | $29.52 \pm 0.08^A$ | 9.14 | 7.56 | 6.88 | 7.95 | $7.88 \pm 1.15^{ABC}$ |
| Oxalate 0.1 mg/L | $54.8 \pm 0.6^{CD}$ | $73.1 \pm 0.4^C$ | 825 | $2.0^A$ | $0.034 \pm 0.001^{AB}$ | $3.38 \pm 1.88^{BC}$ | $27.86 \pm 0.65^{AB}$ | 6.96 | 8.33 | 8.61 | 9.53 | $8.36 \pm 0.95^{ABC}$ |
| Oxalate 500 mg/L | $47.3 \pm 2.3^B$ | $71.6 \pm 0.6^B$ | 720 | $2.5^A$ | $0.039 \pm 0.006^{AB}$ | $2.25 \pm 0.25^C$ | $27.52 \pm 0.65^{AB}$ | 8.01 | 7.50 | 7.27 | 5.68 | $7.12 \pm 1.3$ |
| Tannic Acid 0.1 mg/L | $52.3 \pm 1.2^B$ | $72.7 \pm 0.4^C$ | 930 | $2.0^A$ | $0.03 \pm 0.001^B$ | $5.50 \pm 1.00^{ABC}$ | $27.78 \pm 1.32^{AB}$ | 10.63 | 9.41 | 10.09 | 10.65 | $10.19 \pm 1.05^A$ |
| Tannic Acid 500 mg/L | $54.4 \pm 2.7^{BC}$ | $70.8 \pm 0.3^B$ | 660 | $1.5^A$ | $0.046 \pm 0.005^A$ | $2.50 \pm 0.50^{BC}$ | $29.9 \pm 0.46^A$ | 6.08 | 7.36 | 8.80 | 8.05 | $7.57 \pm 1.44^{BC}$ |
| Organics (SRNOM) 0.1 mg/L | $61.7 \pm 2.6^D$ | $71.1 \pm 0.2^B$ | 870 | $2.5^A$ | $0.030 \pm 0.001^B$ | $9.75 \pm 2.75^A$ | $25.73 \pm 0.59^A$ | 7.54 | 8.31 | 8.30 | 5.73 | $7.47 \pm 1.11^{BC}$ |
| Organics (SRNOM) 10 mg/L | $45.0 \pm 2.6^A$ | $73.4 \pm 0.4^C$ | 975 | $2.0^A$ | $0.030 \pm 0.001^B$ | $5.00 \pm 2.50^{ABC}$ | $29.25 \pm 0.05^B$ | 5.92 | 7.84 | 5.90 | 4.32 | $6.00 \pm 1.35^C$ |

Formatted Table

**Table 4.** Hydrodynamic scaling of finger width measurements resulting from control (NaCl+HNS), plant exudates (citrate and oxalate), and soil components (tannic acid and organics: Suwanee River Natural Organic Matter (SRNOM)) infiltration in ASTM graded sand C778. Finger width is scaled to density, gas-liquid interface tension, density and gas-liquid interface tension of pore water, and $(\cos\theta)^{1/2}$.

| Solution and concentration | | Solution density | Porous media (Graded sand from U.S. silica sand) | Pore water velocity | Average measured finger diameter | Scaling of average measured finger diameter with the solution density relative to pore water | Scaling of average measured finger diameter with the solution gas-liquid interface tension relative to pore water | Scaling of average measured finger diameter with the solution density and gas-liquid interface tension relative to pore water | Scaling of average measured finger diameter with the solution $(\cos\theta)^{1/2}$ relative to pore water |
|---|---|---|---|---|---|---|---|---|---|
| | | (g·cm⁻³) | | (cm·min⁻¹) | (cm) | (cm) | (cm) | (cm) | (cm) |
| NaCl+HSN | | 0.9976 | ASTM C778 | 0.22 | 9.20 | 9.20 | 9.20 | 9.20 | 9.20 |
| Citrate | 0.1 mg/L | 0.9993 | ASTM C778 | 0.22 | 6.60 | 6.59 | 6.17 | 6.16 | 7.84 |
| | 500 mg/L | 0.9968 | ASTM C778 | 0.22 | 7.90 | 7.91 | 7.68 | 7.68 | 10.04 |
| Oxalate | 0.1 mg/L | 0.9972 | ASTM C778 | 0.22 | 8.40 | 8.40 | 8.10 | 8.11 | 9.72 |
| | 500 mg/L | 1.0028 | ASTM C778 | 0.22 | 7.10 | 7.06 | 6.71 | 6.68 | 8.91 |
| Tannic Acid | 0.1 mg/L | 0.9985 | ASTM C778 | 0.22 | 10.20 | 10.19 | 9.79 | 9.78 | 12.16 |
| | 500 mg/L | 1.0002 | ASTM C778 | 0.22 | 7.60 | 7.58 | 7.10 | 7.08 | 8.84 |
| Organics (SRNOM) | 0.1 mg/L | 0.9955 | ASTM C778 | 0.22 | 7.50 | 7.52 | 7.04 | 7.05 | 7.87 |
| | 500 mg/L | 0.9983 | ASTM C778 | 0.22 | 6.00 | 6.00 | 5.81 | 5.81 | 7.69 |
| Average | | | | | 7.83 | 7.83 | 7.51 | 7.51 | 9.14 |
| C.V. | | | | | 1.29 | 1.30 | 1.33 | 1.33 | 1.41 |

Fully Formed Finger Averages / 10 minut Finger Avera

Organics 0.1 / Organics 10 / Tannic Acid 0.1 / Tannic Acid 500 / Oxalate 0.1 / Oxalate 500 / Citrate 0.1 / Citrate 500 / NaCl+HNS / Organics 0.1 / Organics 10 / Tannic Acid 0.1 / Tannic Acid 500 / Oxalate 0.1 / Oxalate 500

61.7 / 45.0 / 52.3 / 54.4 / 54.8 / 47.3 / 52.6 / 46.0 / 64.5 / 61.7 / 45.0 / 52.3 / 54.4 / 54.8 / 47.3
±2.6 / ±2.6 / ±1.2 / ±2.7 / ±0.6 / ±2.3 / ±2.3 / ±4.9 / ±2.6 / ±2.6 / ±1.2 / ±2.7 / ±0.4 / ±0.6 / ±2.3

71.1 / 73.4 / 72.7 / 70.8 / 73.1 / 71.6 / 70.8 / 73.6 / 75.8 / 71.1 / 73.4 / 72.7 / 70.8 / 73.1 / 71.6
±0.2 / ±0.4 / ±0.4 / ±0.3 / ±0.4 / ±0.6 / ±0.4 / ±1.1 / ±0.5 / ±0.2 / ±0.4 / ±0.4 / ±0.3 / ±0.4 / ±0.6

870 / 975 / 930 / 660 / 825 / 720 / 750 / 840 / 945 / 600 / 600 / 600 / 600 / 600 / 600

2.5 / 2 / 2 / 1.5 / 2 / 2.5 / 1.5 / 2 / 4 / 2.5 / 2 / 2 / 2.5 / 2.5 / 2.5

0.03 / 0.03 / 0.03 / 0.05 / 0.03 / 0.04 / 0.04 / 0.04 / 0.03 / 0.03 / 0.02 / 0.03 / 0.04 / 0.03 / 0.04

9.8 / 5.0 / 5.5 / 2.5 / 3.4 / 2.3 / 3.5 / 3.9 / 8.1 / 7.0 / 3.3 / 4.3 / 2.5 / 3.4 / 2.3

25.7 / 29.2 / 27.8 / 29.9 / 27.9 / 27.5 / 29.5 / 29.5 / 28.8 / 17.7 / 13.3 / 19.1 / 25.5 / 20.2 / 22.2

7.5 / 6.0 / 10.2 / 7.6 / 8.4 / 7.1 / 6.6 / 7.9 / 9.2 / 6.6 / 6.0 / 7.8 / 7.5 / 7.4 / 7.0

Formatted [15]
Formatted [16]
[18]
Formatted [19]
Formatted Table [20]
Formatted [21]
Formatted [22]
Formatted [23]
Formatted [24]
Formatted [25]
Formatted [26]

[revised manuscript text omitted]

* * *
**Page 40: [8] Moved to page 40 (Move #5)**                **Author**

| Low | 0.1 | 0.1 | 0.1 | 0.1 |
|-----|-----|-----|-----|-----|
* * *
**Page 40: [9] Deleted**                               **Author**

| Low | 0.1 | 0.1 | 0.1 | 0.1 |
|-----|-----|-----|-----|-----|
* * *
**Page 40: [10] Deleted**                              **Author**
* * *
**Page 40: [11] Deleted**                              **Author**
* * *
**Page 40: [12] Deleted**                              **Author**
* * *
**Page 40: [13] Deleted**                              **Author**

| Group | Solution (mg/L) | Average Contact Angle (θ) | Average Surface Tension (mN/m) | Time (sec) | Average Number of Fingers | Average Velocity (cm/sec) | Average Wetting Depth (cm) | Average Length (cm) | Average Finger Width (cm) |
|---|---|---|---|---|---|---|---|---|---|
| **10 minute Finger Averages** | NaCl+HNS | 64.5 ±2.6 | 75.8 ±0.5 | 600 | 4 | 0.03 | 6.1 | 17.3 | 7.7 |
| | Citrate 500 | 46.0 ±4.9 | 73.6 ±1.1 | 600 | 2 | 0.03 | 2.5 | 19.5 | 7.2 |
| | Citrate 0.1 | 52.6 ±2.3 | 70.8 ±0.4 | 600 | 1.5 | 0.04 | 3.5 | 22.4 | 6.1 |
| | Oxalate 500 | 47.3 ±2.3 | 71.6 ±0.6 | 600 | 2.5 | 0.04 | 2.3 | 22.2 | 7.0 |
| | Oxalate 0.1 | 54.8 ±0.6 | 73.1 ±0.4 | 600 | 2.5 | 0.03 | 3.4 | 20.2 | 7.4 |
| | Tannic Acid 500 | 54.4 ±2.7 | 70.8 ±0.3 | 600 | 2 | 0.04 | 2.5 | 25.5 | 7.5 |
| | Tannic Acid 0.1 | 52.3 ±1.2 | 72.7 ±0.4 | 600 | 2 | 0.03 | 4.3 | 19.1 | 7.8 |
| | Organics 10 | 45.0 ±2.6 | 73.4 ±0.4 | 600 | 2 | 0.02 | 3.3 | 13.3 | 6.0 |
| | Organics 0.1 | 61.7 ±2.6 | 71.1 ±0.2 | 600 | 2.5 | 0.03 | 7.0 | 17.7 | 6.6 |
| **Fully Formed Finger Averages** | Organics 0.1 | 61.7 ±2.6 | 71.1 ±0.2 | 945 | 2.5 | 0.03 | 8.1 | 28.8 | 9.2 |
| | Organics 10 | 45.0 ±2.6 | 73.4 ±0.4 | 840 | 2 | 0.03 | 3.9 | 29.5 | 7.9 |
| | Tannic Acid 0.1 | 52.3 ±1.2 | 72.7 ±0.4 | 750 | 1.5 | 0.05 | 2.5 | 29.5 | 6.6 |
| | Tannic Acid 500 | 54.4 ±2.7 | 70.8 ±0.3 | 720 | 1.5 | 0.05 | 5.5 | 27.5 | 7.1 |
| | Oxalate 0.1 | 54.8 ±0.6 | 73.1 ±0.4 | 825 | 2 | 0.03 | 3.4 | 27.9 | 8.4 |
| | Oxalate 500 | 47.3 ±2.3 | 71.6 ±0.6 | 660 | 2.5 | 0.04 | 2.3 | 29.9 | 7.6 |
| | Citrate 0.1 | 52.6 ±2.3 | 70.8 ±0.4 | 930 | 1.5 | 0.04 | 3.5 | 27.8 | 10.2 |
| | Citrate 500 | 46.0 ±4.9 | 73.6 ±1.1 | 975 | 2 | 0.04 | 5.0 | 29.2 | 6.0 |
| | NaCl+HNS | 64.5 ±2.6 | 75.8 ±0.5 | 870 | 4 | 0.03 | 9.8 | 25.7 | 7.5 |

Section Break (Next Page)

| Page 42: [15] Formatted | Author |
|---|---|

Justified

| Page 42: [16] Formatted | Author |
|---|---|

Font:Bold

| Page 42: [17] Deleted | Author |
|---|---|

 s and

| Page 42: [17] Deleted | Author |
|---|---|

 s and

| Page 42: [18] Deleted | Author |
|---|---|

**Statistical analysis summary table for t-test significant differentiations.**

[revised manuscript text omitted]

---

## Author Response (AR2)

*Editor comments on* "Preferential Flow Systems Amended with Biogeochemical Components: Imaging of a Two-Dimensional Study" *by* Ashley R. Pales et al.

**B. Berkowitz (Editor)**

**10 February 2018**

**Comments to the Author:**

The authors have carefully considered all comments and suggestions, and revised the manuscript accordingly. I believe the revision has led to an improved manuscript, in terms of presentation and clarity. My one concern (reflected also by one referee, in a review of the revised manuscript) is that there are an excessive number of references. I understand that the authors want to be comprehensive in their literature review, but 160 references for a paper that is not a review paper are excessive. For example, recognizing the seminal work of J.Y. Parlange does not need to cite (page 4, line 23) 11 papers! Use "e.g." and choose 2-4 representative papers. The same is true elsewhere.... please reconsider and try to reduce significantly the number of references.

**Response:** The authors thank Editor Brian Berkowitz for his comments. We have revised the manuscript by reducing significantly the number of references.

[revised manuscript text omitted]
 | $64.5 \pm 2.6^A$ | $75.8 \pm 0.5^A$ | 600 | $2.5^C$ | $0.029 \pm 0.003^{AB}$ | $6.13 \pm 2.63^{AB}$ | $17.32 \pm 1.67^{AB}$ | 9.30 | 8.52 | 7.41 | 5.67 | $7.73 \pm 1.$ |
| | Citrate 0.1 mg/L | $52.6 \pm 2.3^D$ | $70.8 \pm 0.4^B$ | 600 | $1.5^B$ | $0.037 \pm 0.003^{AB}$ | $3.50 \pm 1.00^{AB}$ | $22.36 \pm 1.72^{AB}$ | 6.83 | 5.71 | 6.42 | 5.59 | $6.14 \pm 0.$ |
| | Citrate 500 mg/L | $46 \pm 4.9^{BC}$ | $73.6 \pm 1.1^C$ | 600 | $2.0^B$ | $0.032 \pm 0.011^{AB}$ | $2.50 \pm 1.50^{AB}$ | $19.46 \pm 6.63^{AB}$ | 9.23 | 7.56 | 6.68 | 5.50 | $7.24 \pm 1.$ |
| | Oxalate 0.1 mg/L | $54.8 \pm 0.6^{CD}$ | $73.1 \pm 0.4^C$ | 600 | $2.5^B$ | $0.034 \pm 0.005^{AB}$ | $3.38 \pm 1.88^{AB}$ | $20.24 \pm 2.94^{AB}$ | 7.50 | 7.90 | 7.89 | 6.31 | $7.40 \pm 0.$ |
| | Oxalate 500 mg/L | $47.3 \pm 2.3^B$ | $71.6 \pm 0.6^B$ | 600 | $2.5^A$ | $0.037 \pm 0.005^{AB}$ | $2.25 \pm 0.25^B$ | $22.16 \pm 2.88^{AB}$ | 8.41 | 6.55 | 6.77 | 6.13 | $6.97 \pm 0.61$ |
| | Tannic Acid 0.1 mg/L | $52.3 \pm 1.2^B$ | $72.7 \pm 0.4^C$ | 600 | $2.0^B$ | $0.032 \pm 0.003^{AB}$ | $4.25 \pm 0.00^{AB}$ | $19.14 \pm 1.64^{AB}$ | 10.35 | 9.11 | 6.71 | 5.11 | $7.82 \pm 0.32^A$ |
| | Tannic Acid 500 mg/L | $54.4 \pm 2.7^{BC}$ | $70.8 \pm 0.3^B$ | 600 | $2.0^B$ | $0.042 \pm 0.008^A$ | $2.50 \pm 0.50^{AB}$ | $25.49 \pm 4.87^A$ | 5.78 | 6.79 | 8.67 | 8.71 | $7.49 \pm 0.63^A$ |
| | Organics (SRNOM) 0.1 mg/L | $61.7 \pm 2.6^D$ | $71.1 \pm 0.2^B$ | 600 | $2.5^B$ | $0.029 \pm 0.001^{AB}$ | $7.00 \pm 2.00^A$ | $17.66 \pm 0.81^B$ | 7.13 | 7.15 | 7.40 | 4.79 | $6.62 \pm 0.38^A$ |
| | Organics (SRNOM) 10 mg/L | $45.0 \pm 2.6^A$ | $73.4 \pm 0.4^C$ | 600 | $2.0^B$ | $0.022 \pm 0.000^B$ | $3.25 \pm 0.75^{AB}$ | $13.29 \pm 0.15^A$ | 6.30 | 5.49 | 6.31 | 5.89 | $6.00 \pm 0.16^A$ |
| **Fully Formed Finger Average** | NaCl+HNS | $64.5 \pm 2.6^A$ | $75.8 \pm 0.5^A$ | 945 | $2.5^A$ | $0.031 \pm 0.004^B$ | $8.13 \pm 0.63^{AB}$ | $28.84 \pm 1.25^A$ | 10.05 | 9.06 | 9.53 | 8.30 | $9.24 \pm 0.84^{AB}$ |
| | Citrate 0.1 mg/L | $52.6 \pm 2.3^D$ | $70.8 \pm 0.4^B$ | 750 | $1.5^A$ | $0.039 \pm 0.002^{AB}$ | $3.50 \pm 1.00^{BC}$ | $29.46 \pm 0.64^A$ | 6.99 | 6.75 | 6.87 | 5.93 | $6.63 \pm 0.67^C$ |
| | Citrate 500 mg/L | $46 \pm 4.9^{BC}$ | $73.6 \pm 1.1^C$ | 840 | $2.0^A$ | $0.037 \pm 0.008^{AB}$ | $3.88 \pm 2.88^{BC}$ | $29.52 \pm 0.08^A$ | 9.14 | 7.56 | 6.88 | 7.95 | $7.88 \pm 1.15^{ABC}$ |
| | Oxalate 0.1 mg/L | $54.8 \pm 0.6^{CD}$ | $73.1 \pm 0.4^C$ | 825 | $2.0^A$ | $0.034 \pm 0.001^{AB}$ | $3.38 \pm 1.88^{BC}$ | $27.86 \pm 0.65^{AB}$ | 6.96 | 8.33 | 8.61 | 9.53 | $8.36 \pm 0.95^{ABC}$ |
| | Oxalate 500 mg/L | $47.3 \pm 2.3^B$ | $71.6 \pm 0.6^B$ | 720 | $2.5^A$ | $0.039 \pm 0.006^{AB}$ | $2.25 \pm 0.25^C$ | $27.52 \pm 0.65^{AB}$ | 8.01 | 7.50 | 7.27 | 5.68 | $7.12 \pm 1.32^{BC}$ |
| | Tannic Acid 0.1 mg/L | $52.3 \pm 1.2^B$ | $72.7 \pm 0.4^C$ | 930 | $2.0^A$ | $0.03 \pm 0.001^B$ | $5.50 \pm 1.00^{ABC}$ | $27.78 \pm 1.32^{AB}$ | 10.63 | 9.41 | 10.09 | 10.65 | $10.19 \pm 1.05^A$ |
| | Tannic Acid 500 mg/L | $54.4 \pm 2.7^{BC}$ | $70.8 \pm 0.3^B$ | 660 | $1.5^A$ | $0.046 \pm 0.005^A$ | $2.50 \pm 0.50^{BC}$ | $29.9 \pm 0.46^A$ | 6.08 | 7.36 | 8.80 | 8.05 | $7.57 \pm 1.44^{BC}$ |
| | Organics (SRNOM) 0.1 mg/L | $61.7 \pm 2.6^D$ | $71.1 \pm 0.2^B$ | 870 | $2.5^A$ | $0.030 \pm 0.001^B$ | $9.75 \pm 2.75^A$ | $25.73 \pm 0.59^A$ | 7.54 | 8.31 | 8.30 | 5.73 | $7.47 \pm 1.11^{BC}$ |
| | Organics (SRNOM) 10 mg/L | $45.0 \pm 2.6^A$ | $73.4 \pm 0.4^C$ | 975 | $2.0^A$ | $0.030 \pm 0.001^B$ | $5.00 \pm 2.50^{ABC}$ | $29.25 \pm 0.05^B$ | 5.92 | 7.84 | 5.90 | 4.32 | $6.00 \pm 1.35^
[revised manuscript text omitted]

| Page 34: [7] Deleted | Author |
| --- | --- |

**Page 34: [8] Deleted**                              **Author**

**Page 35: [9] Deleted**                              **Author**

**Page 35: [10] Deleted**                             **Author**